*Resource*

# The TOR-dependent phosphoproteome and regulation of cellular protein synthesis

Tiffany Mak[1,†,*] , Andrew W Jones[1,2] & Paul Nurse[1,3]

## Abstract

Cell growth is orchestrated by a number of interlinking cellular processes. Components of the TOR pathway have been proposed as potential regulators of cell growth, but little is known about their immediate effects on protein synthesis in response to TOR-dependent growth inhibition. Here, we present a resource providing an in-depth characterisation of *Schizosaccharomyces pombe* phosphoproteome in relation to changes observed in global cellular protein synthesis upon TOR inhibition. We find that after TOR inhibition, the rate of protein synthesis is rapidly reduced and that notable phosphorylation changes are observed in proteins involved in a range of cellular processes. We show that this reduction in protein synthesis rates upon TOR inhibition is not dependent on S6K activity, but is partially dependent on the *S. pombe* homologue of eIF4G, Tif471. Our study demonstrates the impact of TOR-dependent phospho-regulation on the rate of protein synthesis and establishes a foundational resource for further investigation of additional TOR-regulated targets both in fission yeast and other eukaryotes.

**Keywords** phosphoproteomics; protein synthesis; TOR regulation

**Subject Categories** Metabolism; Post-translational Modifications & Proteolysis; Signal Transduction

**The EMBO Journal (2021) 40: e107911**

## Introduction

Cell growth, the accumulation of biomass and cell reproduction, underpin the overall growth of all living organisms. Biomass is made up of cellular components and macromolecules, a major constituent of which is protein. Protein biosynthesis is considered the largest energy-consuming process in actively growing cells, accounting for ~30% in mammalian cells and ~50% in bacterial cells of total energy consumption (Buttgereit & Brand, 1995; Russell & Cook, 1995). The control of the global cellular rate of protein synthesis is therefore central to understanding cellular growth control. In eukaryotic cells, the target of rapamycin (TOR) signalling pathway is thought to play a key role in regulating cell growth. TOR is a serine/threonine protein kinase that is a member of the phosphatidylinositol kinase-related kinase (PIKK) family. It was discovered in budding yeast through characterising mutants resistant to the growth inhibitor rapamycin (Heitman *et al*, 1991) and is conserved across eukaryotes (Brown *et al*, 1994). The TOR kinases form two protein complexes, TOR complex 1 (TORC1) and TOR complex 2 (TORC2). Studies in both yeasts and mammalian cells have suggested TORC1 as the major complex involved in cellular growth control (Loewith *et al*, 2002; Wullschleger *et al*, 2006; Weisman *et al*, 2007; Yang *et al*, 2013). Despite the large body of work that has been undertaken to characterise TOR-regulated processes, surprisingly little is known about how components downstream of TOR signalling are involved in altering the rates of protein synthesis. Therefore, to understand eukaryotic growth control, it is critical to elucidate how TOR-regulated pathways modulate growth processes, particularly the rate of global cellular protein synthesis.

Previous studies have focussed on the cellular effects of inhibiting the TOR kinases. Rapamycin treatment leads to a progressive decrease in the rate of protein synthesis in budding yeast (Barbet *et al*, 1996) and to a lesser extent in mammalian cells (Beretta *et al*, 1996). More potent and selective inhibitors of TOR, such as Torin1, specifically target the catalytic site of TOR (Thoreen *et al*, 2009), inhibiting cellular growth in both mammalian (Thoreen *et al*, 2012) and fission yeast (Atkin *et al*, 2014) cells. Torin1 inhibits the function of both TOR kinases in fission yeast, which are found in two distinct protein complexes, TORC1 (contains the Tor2 kinase) and TORC2 (contains the Tor1 kinase) (Weisman *et al*, 2007). Inhibition of growth with Torin1 is also reversible and does not induce cell death even after 24 h of incubation (Atkin *et al*, 2014) and thus provides a useful means to perturb TOR signalling and enable physiological studies upon inhibition of TOR activity.

As TOR and other components within the TOR-signalling pathway are protein kinases, changes in protein phosphorylation have been investigated to identify potential targets involved in regulating cellular growth processes after TOR inhibition. The S6 kinases and translation initiation-related factors, including the 4E-binding protein (4E-BP) and eIF4F complex, are among the most widely discussed proteins as potential downstream targets of TOR in having

1 Cell Cycle Laboratory, The Francis Crick Institute, London, UK
2 Protein Analysis and Proteomics Platform, The Francis Crick Institute, London, UK
3 Laboratory of Yeast Genetics and Cell Biology, Rockefeller University, New York, NY, USA
*Corresponding author (lead contact). Tel: +4593511736; E-mail: tifmak@biosustain.dtu.dk
†Present address: The Novo Nordisk Foundation Center for Biosustainability, Technical University of Denmark, Kgs. Lyngby, Denmark

an impact on protein synthesis upon inhibition (Schmelzle & Hall, 2000; Averous & Proud, 2006; De Virgilio & Loewith, 2006; Ma & Blenis, 2009). Early studies in budding yeast suggested that Sch9 (a S6K orthologue) is a major target of TORC1 and is thought to regulate translation initiation through transcriptional regulation of RNA pol III and pol I-dependent genes, many of which are ribosome biogenesis (ribi)-related genes (Urban *et al*, 2007); (Huber *et al*, 2009). However, studies in mice have shown that a complete knock-out of S6K1 did not affect global translational activity (Garelick *et al*, 2013), raising questions about the role of the S6 kinases and their downstream phosphorylation targets in regulating translational activity.

Phosphoproteomic studies undertaken in both budding yeast and mammalian cells have uncovered interactions between TOR and various pathways that regulate growth, including the PKA (Soulard *et al*, 2010) and insulin signalling pathways (Hsu *et al*, 2011; Yu *et al*, 2011), as well as pyrimidine and amino acid biosynthesis (Robitaille *et al*, 2013; Oliveira *et al*, 2015). However, it still remains unclear how these observed phosphoproteomic changes act to influence cellular growth or protein synthesis. Most of these phosphorylation studies have lacked sufficient temporal resolution to correlate phosphosphorylation changes with alterations in growth processes. Their phosphosite analyses were also limited by the mass spectrometry techniques employed. Recent advancements have increased the number of samples that can be processed in parallel, as well as detection sensitivity and reliability, enhancing the molecular resolution and improving the quantitative phosphoproteomic analyses considerably.

Using these improved techniques, we have carried out an in-depth phosphoproteomics study in fission yeast to characterise the changes at fine temporal resolution after inhibition of TOR kinase activity with Torin1. Our study has generated a database of proteins and phosphosite targets potentially involved in modulating protein synthesis and other growth-related processes. Functional categorisation of these proteins, based on cellular processes, provides a more comprehensive overview of the essential functions that may be involved in coordinating growth control. Many of the target proteins identified are conserved in both budding yeast and humans, extending the relevance of the database. By monitoring protein synthesis rates in parallel with TOR inhibition, we are able to correlate the phosphorylation changes with the kinetics of protein synthesis inhibition. This an improved approach to identify potential targets that

may have roles in the regulation of protein synthesis, and are dependent on the TOR-signalling pathway.

## Results

### TOR activity is a major regulator of cellular protein synthesis rates

The rate of global cellular protein synthesis was monitored by measuring the rate of incorporation of the methionine analogue L-homoproparylglycine (HPG) (Knutsen *et al*, 2015; Green & Pelkmans, 2016), after inhibition of TOR with Torin1 in fission yeast. The methionine analogue was tagged with a fluorescent label using click chemistry, with the signal detected at a single-cell level using flow cytometry. Fluorescent signal measured in a population of exponentially growing fission yeast cells was normally distributed on a logs-cale (Fig 1A), and the median rate of analogue incorporation, after an initial lag of 5 min, was linear over 60 min (Fig 1B). Therefore to ensure that the incorporation of the analogue was within the linear range, a pulse of 15 min or longer was used to calculate the rate of protein synthesis (detailed in Materials and Methods).

The selective ATP-competitive inhibitor Torin1 completely inhibits colony formation at 5 μM (Lie *et al*, 2018). We demonstrated that when it was added to exponentially growing fission yeast cells in liquid culture, growth as measured by increase in optical density ($OD_{595}$) began to reduce by 80–90 min, with the rate of mass doubling reducing to a third compared to cells treated with the DMSO solvent alone (Fig 1C). This experiment was repeated using the Torin1-resistant, *tor2*-G2040D mutant (Atkin *et al*, 2014), and the rate of mass doubling was found to be unchanged after Torin1 addition (Fig 1D). This established that the mutant was resistant to inhibitory growth effects of the drug in terms of cellular mass increase and was subsequently used to control for off-target effects of the Torin1 inhibitor.

To investigate the effects of TOR inhibition on protein synthesis, the rate of synthesis was monitored in cells treated with Torin1 at 5 μM for differing lengths of time (Fig 1E). In wild type, the synthesis rate fell to 50% of the starting level within 10 min of Torin1 addition, 75% by 20 min and 90% by 40 min. This shows that the majority of protein synthesis is dependent upon TOR activity, and therefore, we would expect that the activity of potential TOR

**Figure 1.  TOR activity is a major regulator of cellular protein synthesis rates.**

Measuring active protein synthesis using the methionine analogue, L-homopropargylglycine (HPG) and measuring the change in rates of mass doubling and protein synthesis upon inhibition of TOR activity.

A    Density plot showing the measured fluorescence intensity distribution of individual cells for each time point after HPG addition. Time is represented on the y-axis, with the bottom of the distribution plot aligning to the corresponding time point.

B    Median fluorescence intensity of the distribution plots in A plotted against the indicated time after HPG addition. Linear regression line fitted from 5 min onwards.

C, D  Growth rates as measured by the change in optical density ($OD_{595}$) over time for wild type (C) and Torin1-resistant (D) cells upon 5 μM of Torin1 treatment (at $T = 0$ min). Respective strains were treated with the DMSO vehicle alone to control for potential effects of the solvent on cell growth. Logarithmic values of the $OD_{595}$ increase are plotted against time in minutes. Data aggregated from three independent experiments with error bars representing the standard deviations (SD). Linear regressions were fitted using the curve-fitting function on Prism9. Ratios in the rate of mass doubling (as calculated from gradient of regression lines detailed in Materials and Methods) of Torin1-treated cells to DMSO control were 0.34 in wild type (C) and 0.97 for Torin1-resistant (D) cells.

E    Rate of protein synthesis as measured by the amount of HPG incorporation after 20 min of differing time intervals (0–60 min) of 5 μM Torin1 for wild type (green) and *tor1Δ* (yellow). Values are normalised to $T = 0$ min. Exponential one-phase decay non-linear regressions were fitted using the curve-fitting function on Prism9. Data aggregated from 3 independent experiments with error bars representing the standard deviations (SD).

F    Growth rates for wild type and *tor1Δ*, as measured by the change in optical density ($OD_{595}$) over time. Linear regressions fitted with Prism9.

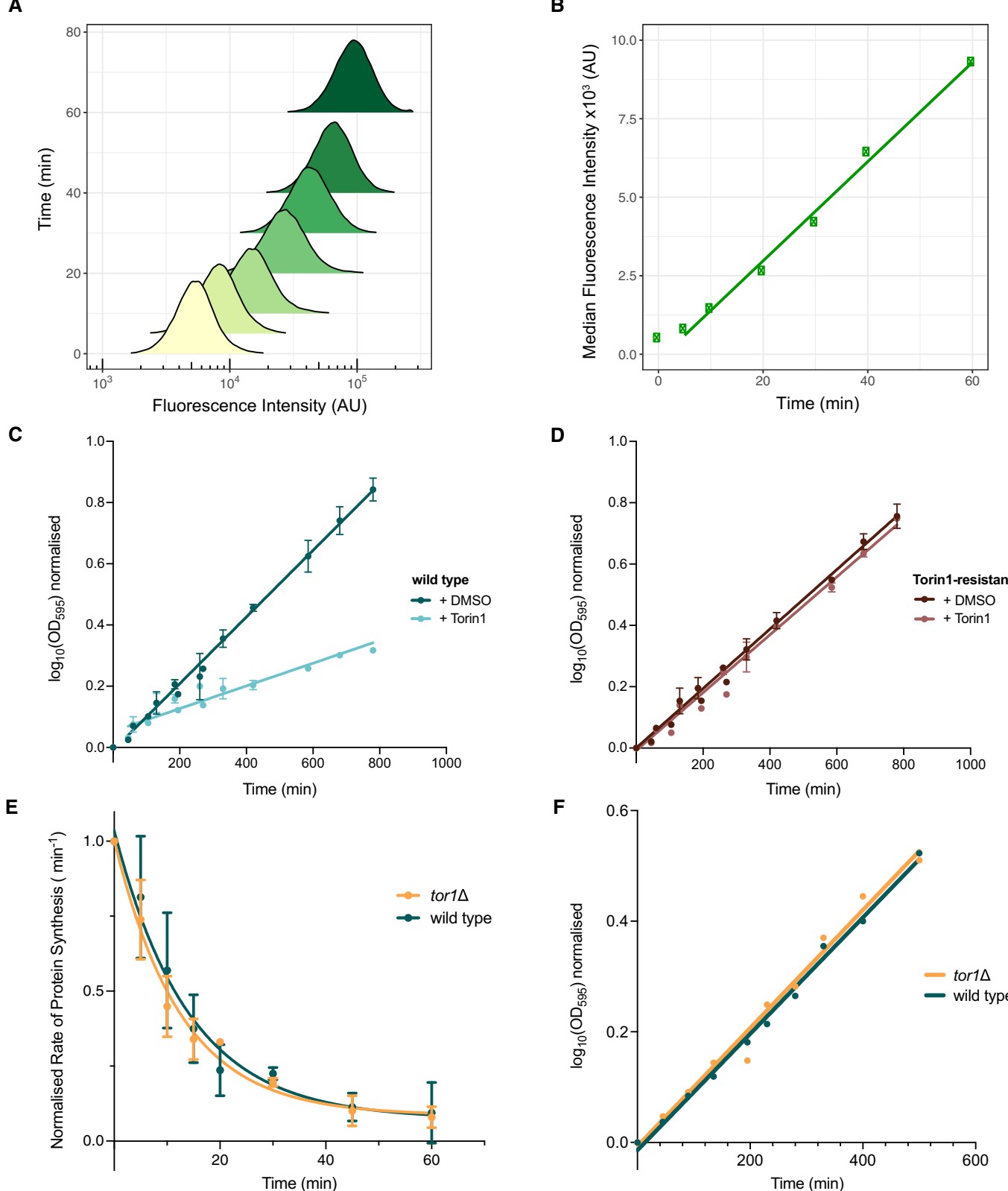

**Figure 1.**

regulatory targets would also be altered within 10–20 min. We next assayed the protein synthesis rate in a *tor1Δ* mutant strain, which eliminates the non-essential TORC2. There was no measurable difference in the growth rates between wild type and *tor1Δ* cells (Fig 1F), and the changes in rates of protein synthesis upon TOR inhibition were identical (Fig 1E). These observations are consistent with the idea that the essential TORC1 is the major complex responsible for the rapid inhibition of protein synthesis and that changes in phosphorylation levels relevant to regulating protein synthesis should be detectable within 20 min and significant by 40 min of TOR inhibition.

### TOR-dependent phosphoproteomic changes identify conserved proteins with potential roles in modulating rates of protein synthesis

Based on these results, we carried out a proteomics and phosphoproteomics time course study to identify the molecular changes responsible for the reduction in rates of biomass accumulation and protein synthesis after TOR inhibition. Exponentially growing wild type and Torin1-resistant cells were treated with Torin1 (5 μM) and sampled at intervals from 5 to 85 min. The samples were labelled with isobaric tandem mass tag labels (TMT10plex) for relative quantification of both the protein and individual phospho-peptide levels (detailed in Materials and Methods).

A total of 3,467 proteins were identified across all samples, accounting for ~70% of the fission yeast proteome (Table EV1). After controlling for off-target effects of DMSO (the Torin1 solvent; see Materials and Methods), none of the proteins showed more than a 2-fold change in protein levels relative to the wild type starting levels ($T = 0$ min), in either wild type or Torin1-resistant cells, after 40 min of Torin1 treatment (Fig 2A), when the rate of protein synthesis was reduced by around 90% (Fig 1E). A closer look at the frequency distribution of the changes in wild type cells showed that 98% of the proteins (3,323/3,393) were within a ± 1.3-fold range for their changes in protein levels (Fig 2B). And for those that showed changes between 1.3 to 2-fold, corresponding changes were also observed in Torin1-resistant cells, as indicated by the positively correlated values on Fig 2A (y = x in light grey, plotted as guide), suggesting that these are likely due to non-specific and off-target effects of DMSO or Torin1 treatment. The combined observations in Fig 2A and B therefore indicate that the changes in protein levels are unlikely to play a role in regulating the rate of protein synthesis after TOR inhibition.

We next analysed the phosphoproteome following TOR inhibition. To confirm the effectiveness of Torin1 inhibition, we first analysed the phosphorylation behaviour of the 40S ribosomal protein S6 (RPS6), a well-established biochemical read-out of TORC1 and TORC2 activity (Nakashima *et al*, 2010; Du *et al*, 2012; Atkin *et al*, 2014). In fission yeast, there are two copies of the gene encoding for the Rps6 protein, *rps601* and *rps602*, each containing two well-conserved phosphosites at positions S235 and S236, both of which were identified in our study. In both proteins, the two phosphosites exhibited rapid dephosphorylation kinetics in the respective proteins, exceeding close to 4-fold within 20 min of Torin1 treatment in wild type cells (Fig 2C and D). For the Torin1-resistant mutant, the phosphorylation levels remained constant throughout the time course, demonstrating that the dephosphorylation observed in Rps6 phosphosites in wild type cells is from the inhibition of TOR kinases activity, and not due to off-target effects of Torin1 treatment. Further information can be inferred from the multiplicity (denoted by the M-number in brackets) of the two phosphosites on each of the proteins. In this case, M2 indicates that two phosphorylation events were detected within the same fragmented peptide at the same time phosphorylation was detected on the specified phosphosite. We also detected the M1 versions of the S235 and S236 phosphosites in this experiment, but the phosphorylation changes were much less immediate than the M2 phosphosites (Fig 2E and F). This suggests that their phosphorylation behaviours are likely dependent on each other, possibly cooperative.

This phosphorylation study identified a total of 20,072 phosphosites on 2,271 phosphoproteins, over 3 times the number of phosphosites identified from previous TOR phosphoproteomics studies in both budding yeast and mammalian cells. We controlled for non-specific and off-target phosphorylation changes caused by DMSO or Torin1 treatment by taking into account the phosphorylation values of the Torin1-resistant strain at each time point (detailed in Materials and Methods, Table EV2). A total of 1,043 phosphosites were identified to show a more than 2-fold increase or decrease in phosphorylation at the end of the 85-min time course, accounting for 5% of the overall detected phosphosites. Of these, 650 phosphosites on 264 phosphoproteins exhibited a more than 2-fold phosphorylation change relative to starting levels within 40 min of Torin1 addition (Table EV3). Among the 264 phosphoproteins, 160 have homologues in both budding yeast and human cells, forming a database of proteins under immediate phosphorylation control upon TOR kinase inhibition (conservation indicated on Table EV3). An accompanying data visualisation tool (Shiny app) has been made available

---

**Figure 2.  TOR-dependent phosphoproteomic changes identifies conserved proteins with potential roles in modulating rates of protein synthesis.**  ▶

Investigating the protein expression levels for the phosphoproteomics time course study and validation of experimental conditions based on known indirect TOR phosphorylation targets.

A    Scatter plot showing the protein levels in wild type cells after 40 min of Torin1 (5 μM) addition plotted against the corresponding values for the same protein in Torin1-resistant cells after 40 min. Dark grey dashed lines at x = ±1 indicate the 2-fold threshold, of which no proteins exceeded in the study. The light grey dashed line represents y = x, plotted as guide to indicate positive correlation in protein levels between wild type and Torin1-resistant cells.

B    Frequency distribution of the proteins levels after 40 min of Torin1 treatment in wild type cells (*n* = 3,380), with 98% of all proteins (3,314/3,380) within a 1.3-fold boundary, as indicated by the grey dashed lines at ± 0.38.

C–F  Phosphorylation changes of the conserved phosphosites, S235 and S236, on Rps601(C) and Rps602 (D–F), over 85 min of Torin1 (5 μM) treatment wild type (green) and Torin1-resistant (brown) cells. M-values (in brackets) represent the different multiplicities of the same phosphosites. Lines were fitted with the curve-fitting function on Prism9 using the non-linear regressions exponential one-phase decay (C, D) or plateau followed by one-phase decay (E, F) functions, respectively. Grey dashed lines indicate 2-fold threshold.

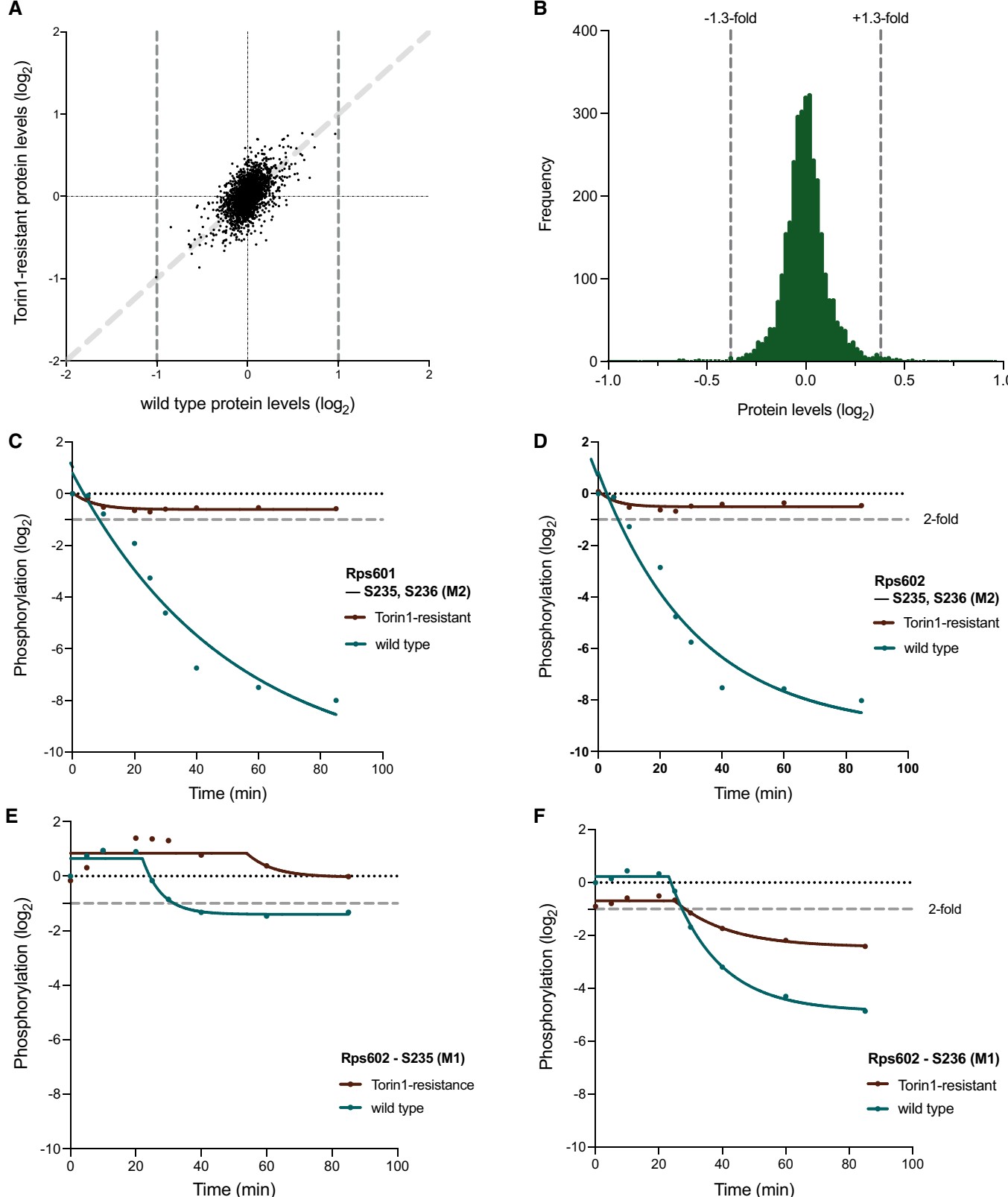

**Figure 2.**

Table 1.   17 major GO Biological Process categories for the 160 conserved proteins with phosphosites showing more than 2-fold change within 40 min of Torin1 addition.

| GO term | Frequency | Gene names (Essential genes in bold) |
|---|---|---|
| Signalling (GO:0023052) | 37 (**5**) | amk2, atg1, bzz1, cek1, far11, gtr2, igo1, iml1, inp53, **kcs1**, **mor2**, **orb6**, pab1, par1, **pik1**, pom1, pom2, ppk19, ppk25, ppk30, ras1, rga3, rgf1, rnc1, **rng2**, sat1, sck1, sck2, sea2, sea3, sec73, SPAC824.09c, SPBC16E9.02c, stm1, syt22, win1, wis1 |
| Vesicle-Mediated Transport (GO:0016192) | 33 (**10**) | bzz1, cdc15, ede1, gga21, **glo3**, gyp1, irs4, mpd2, mup1, nxt3, **orb6**, osh2, pan1, pep12, **pob1**, ppk19, sat1, **sec16**, **sec2**, **sec27**, sec31, **sec5**, sec73, **sec9**, SPAC824.09c, SPCC622.14, sst4, **syb1**, syt22, trs130, ups17, **uti1**, wdr44 |
| Cytoplasmic Translation (GO:0002181) | 15 (**10**) | **erf1**, **hrs1**, **rpl8**, rps101, rps601, rps602, slh1, **SPAC29E6.06c**, sum2, **sum3**, **sup35**, **tif212**, **tif225**, **tif303**, **tif471** |
| Ribosome Biogenesis (GO:0042254) | 15 (**4**) | **enp1**, noc12, nup124, **nup146**, nup40, pxr1, rei1, **rpl8**, rps101, rps601, rps602, SPBC16H5.08c, SPBC4.02c, **tif471**, tif6 |
| Actin Cytoskeleton Organization (GO:0030036) | 13 (**2**) | bud6, bzz1, cdc15, ede1, imp2, myp2, **orb6**, pan1, pom1, ppk30, pxl1, **rng2**, tea4 |
| Autophagy (GO:0006914) | 13 (**1**) | atg1, atg11, atg13, atg3, atg9, gtr2, iml1, irs4, nxt3, osh2, ppk19, **sec16**, ubx3 |
| Establishment or Maintenance of Cell Polarity (GO:0007163) | 13 (**2**) | alp14, bud6, cdc15, **mor2**, mto1, **orb6**, osh2, pom1, ras1, syt22, tea1, tea3, tea4 |
| Chromatin Organization (GO:0006325) | 12 (**2**) | bdf2, eaf6, eaf7, hta2, nap1, nap2, **pst3**, set2, sgf29, shf1, SPBC4.02c, **taf12** |
| Membrane Organization (GO:0061024) | 12 (**6**) | cdc15, **glo3**, mup1, **ned1**, pep12, **sec16**, sec31, **sec9**, **syb1**, ubx3, ups17, **uti1** |
| Protein-Containing Complex Assembly (GO:0065003) | 12 (**6**) | alp14, **brr2**, bud6, mto1, nap1, nap2, **rng2**, **sec16**, **taf12**, **tif212**, **tif303**, tif6 |
| Transcription, DNA-Templated (GO:0006351) | 12 (**6**) | eaf7, **hsf1**, maf1, nup40, **pst3**, **rpc37**, set2, sfp1, sgf29, **taf12**, **tfa1**, **tfg1** |
| Nucleocytoplasmic Transport (GO:0006913) | 11 (**2**) | kap123, **ned1**, npp106, nup124, **nup146**, nup40, nup61, ras1, rei1, SPBC16H5.08c, tif6 |
| mRNA Metabolic Process (GO:0016071) | 9 (**1**) | **brr2**, ccr4, cip1, cip2, not2, not3, rnc1, sts5, vts1 |
| DNA Repair (GO:0006281) | 8 (**2**) | eaf6, eaf7, hta1, hta2, pms1, **pst3**, **rad21**, ung1 |
| Mitotic Cell Cycle Phase Transition (GO:0044772) | 8 (0) | cek1, igo1, pab1, par1, pom1, rad25, sck2, wis1 |
| Mitotic Cytokinesis (GO:0000281) | 8 (**1**) | cdc15, igo1, imp2, myp2, pom1, pom2, pxl1, **rng2** |
| Transmembrane Transport (GO:0055085) | 8 (**1**) | abc2, avt3, can1, pmc1, sbh1, **sec62**, SPCC1672.11c, stm1 |
| Uncategorised | 17 (0) | ada1, dml1, gde1, gpd1, izh3, knk1, nrp1, ntp1, nuj2, pcc1, pcy1, scw1, SPAC22A12.14c, SPBC30D10.05c, tcb2, tmf1, ubp7 |

online for visualising the individual phosphorylation kinetics (https://tmak.shinyapps.io/TM_App-Table_EV3/), and an illustrated example of the workflow is provided in Fig EV1 and detailed in Materials and Methods.

**Cellular coordination of growth control**

To gain an overview of the proteins involved in the coordination of cellular growth control, we looked to see what key functional processes are represented among the proteins that underwent rapid phosphorylation changes upon TOR kinase inhibition. Since the TOR-signalling pathway and its interacting network are highly conserved across eukaryotes, we focussed on the 160 conserved proteins containing homologues in both human cells and budding yeast. We classified the proteins based on their Gene Ontology (GO) Biological Process annotations (from PomBase Lock *et al*, 2019) and

found that 90% of the proteins (143/160) were represented by 17 major categories (Table 1). As it is likely that the core regulatory proteins needed to regulate growth are also required for cellular survival, we looked first to see which of the proteins are encoded by essential genes. Of the conserved proteins, 23% (36/160) were found to be essential for cell viability (Table EV3), a third of which (12/36) have roles in cytoplasmic translation (GO:0002181) and ribosome biogenesis (GO:0042254). This supports the view that translation is a major point of regulation for TOR-dependent growth control.

Next, we performed a GO enrichment analysis on the 160 conserved proteins to identify other cellular processes that are potentially involved in TOR-dependent growth control (Table EV4). We found that proteins responsible for establishing and maintaining cell polarity (GO:0061246) were among the most highly enriched categories (14.5-fold). Other structural organisation-related

categories were also seen to be enriched, with proteins involved in cortical actin cytoskeleton organisation (GO:0030865) and organelle localisation (GO:0051640), exhibiting a 4.2- and 2.9-fold enrichment, respectively (Table EV4). This highlights that the structural organisation of a cell is heavily regulated upon TOR inhibition.

As TOR is a major signalling kinase within the cell, it is unsurprising that many signalling-related proteins are heavily regulated upon TOR inhibition. Overall, signalling-related proteins (GO: 0023052) account for the largest group of proteins (37/160) that exhibit rapid phosphorylation changes upon TOR inhibition. In particular, enrichment analysis identified a 5.6-fold enrichment for those involved in the negative regulation of intracellular signal transduction (GO:1902532; Table EV4). Other notable categories include proteins responsible for retrograde transport from the endosome to Golgi (GO:0042147; 7.6-fold), which are partly represented by vesicle-mediated transport proteins (GO:0016192; 2.5-fold), as well as those involved in autophagy-related processes (G0:0000422; 9.9-fold and GO:0000045; 9.1-fold) (Table EV4). These observations provide evidence that TOR-dependent growth regulation is highly coordinated and involves multiple major cellular processes.

## TORC1-dependent phosphorylation regulates the rate of protein synthesis

Given that the *S. pombe* TORC2 (containing Tor1) is not involved in the rapid inhibition of protein synthesis (Fig 1E), we focussed on TORC1, the complex containing the essential Tor2 kinase. Using similar proteomic and phosphoproteomic approaches, we investigated the relative phosphorylation changes in wild type and *tor1Δ* cells to focus on potential targets that could play a critical role in modulating the rates of protein synthesis following TOR inhibition. We performed a shorter 40-min time course experiment, by which time the rate of protein synthesis is much reduced, and sampled at higher frequency to gain a better temporal resolution of the protein and phosphorylation changes occurring at early time points following Torin1 addition (see Materials and Methods). This second time course largely replicates the critical part of the first time course, and examination of the same phosphosites from the two experiments indicates that the data are reproducible.

At the protein level, 3707 proteins were detected in both the wild type and *tor1Δ* cells across all time points (Table EV1). None of the detected proteins showed more than a 2-fold change in expression levels in either of the two strains after 40 min of Torin1 treatment (Fig EV2). We investigated whether there were inherent differences in the protein levels between cells containing both functional TOR complexes (wild type) or only the essential TOR complex 1 (*tor1Δ*). The corresponding levels of individual proteins in *tor1Δ* cells were compared to the levels measured in wild type cells before background subtraction and before Torin1 addition. 26 proteins showed over a 1.5-fold difference in expression in *tor1Δ* compared to wild type cells before Torin1 treatment (Protein IDs in red in Table EV1 and red in Fig EV2), indicating that the expression of these proteins is affected by the absence of TORC2 during steady-state growth. Inhibition of TOR activity, however, did not cause a change in relative expression of these proteins when compared to their background expression levels in the respective strains, suggesting that the absence of the non-essential Tor1 kinase does not have an effect on protein expression upon inhibition of TOR activity (highlighted in red in Fig EV2). The observation of unchanged protein expression upon TOR inhibition in either of the two strains supports our earlier conclusion that changes in the rates of protein synthesis upon TOR inhibition are unlikely to involve changes in protein levels.

A total of 7,162 phosphosites were identified across all 20 samples in this experiment, representing a total of 1,532 phosphoproteins, 89% of which (6,340/7,162) were also in common with the 85-min time course (Table EV2). 372 phosphosites on 177 proteins showed a more than 2-fold change within 40 min of Torin1 treatment in wild type cells (marked on Table EV3). To focus on the changes that are dependent upon TORC1, we first excluded phosphosites that were dependent on TORC2. 43 of the 374 sites showed a more than 2-fold change in phosphorylation in wild type cells, but remained below a 1.3-fold difference in phosphorylation in *tor1Δ* cells within 40 min of Torin1 addition (Fig 3A and B; marked as "TORC2" on Table EV3 and visualisable on Shiny App, see Materials and Methods). This indicates that the phosphorylation changes detected in wild type cells are dependent upon the presence of the non-essential TORC2 complex and hence were excluded from further analysis.

Following exclusion of the TORC2-dependent sites, we analysed the remaining phosphosites and identified 158 phosphosites on 83 proteins that showed a more than 2-fold change for both wild type and *tor1Δ* within 40 min (Fig 3C and D; marked as "TORC1" on

**Figure 3. Identifying a core TORC1-dependent protein interaction network in regulating rates of protein synthesis.**

Identification of a TORC1-dependent protein interaction network through comparative phosphoproteomic studies.

A–D   Summary profile plots illustrating the minimum and maximum range of phosphorylation for the respective phosphorylation sites in wild type (in green) and *tor1Δ* (in yellow) cells, with lines indicating the median values of the phosphorylation profiles and the shaded areas representing the range of values. Values are normalised to their respective values at $T = 0$ min in each strain. Top two panels represent the 43 TORC2-dependent phosphosites showing more than 2-fold change in wild type (A) but within 1.3-fold difference in *tor1Δ* cells (B). Bottom two panels illustrate the 158 phosphosites showing relative phosphorylation changes of more than 2-fold in both wild type (C) and *tor1Δ* (D) cells.

E   Interaction network generated from the interactome analysis of the 49 potential TORC1-dependent proteins conserved among fission yeast, budding yeast and human, based on collated annotations from the STRING database. Interactions between proteins were scored based on both functional and physical interactions from experimental, text mining and database sources only. Connections were drawn upon reaching a minimum interaction score of 0.4, and the lines were weighted to indicate the relative strength of the annotated interactions. The resultant network of the 28 connected proteins were functionally categorised based on their GO biological processes as annotated on PomBase and represented by the following colours, respectively: cytoplasmic translation (GO:0002181) and ribosome biogenesis (GO:0042254) in green, mRNA metabolism (GO:0016071) and transcription (GO:0006351) in purple, vesicle-mediated transport (GO:0016192) in yellow and actin cytoskeleton organisation (GO:0030036) in brown. Grey nodes indicate proteins that were not assigned to the highlighted functional categories. Proteins that were identified as common in both phosphoproteomics studies are denoted by the darker hues of the respective colour categories, and the nodes of essential proteins are outlined in white (9/28).

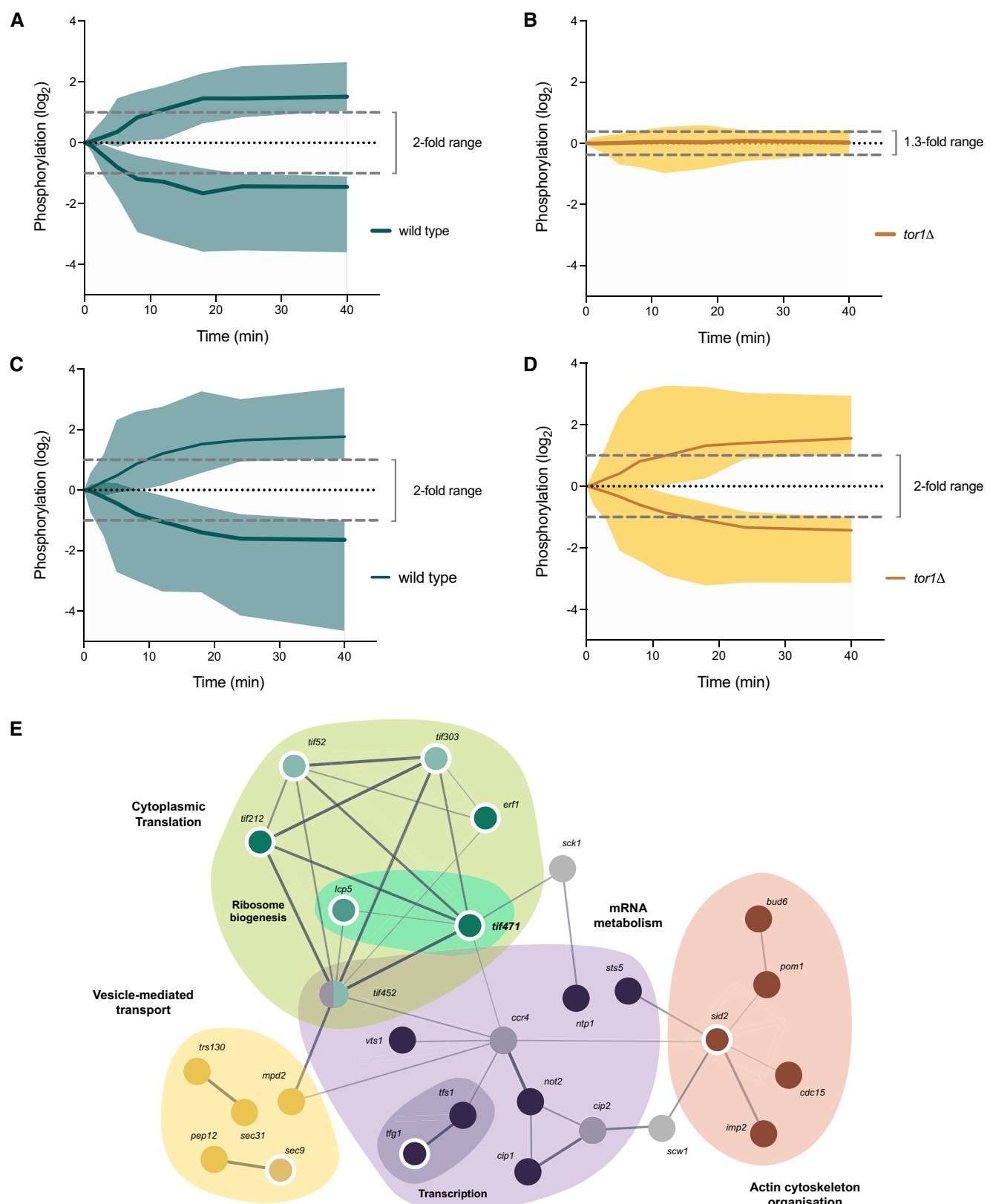

**Figure 3.**

Table 2. All GO biological process categories for the 28 conserved interacting proteins.

| GO term | Frequency | Gene names |
|---|---|---|
| mRNA Metabolic Process (GO:0016071) | 8 | ccr4, cip1, cip2, not2, sts5, tfs1, tif452, uts1 |
| Cytoplasmic Translation (GO:0002181) | 6 | erf1, tif212, tif303, tif452, tif471, tif52 |
| Vesicle-Mediated Transport (GO:0016192) | 6 | cdc15, mpd2, pep12, sec31, sec9, trs130 |
| Actin Cytoskeleton Organization (GO:0030036) | 5 | bud6, cdc15, imp2, pom1, sid2 |
| Establishment/ Maintenance of Cell Polarity (GO:0007163) | 4 | bud6, cdc15, pom1, sid2 |
| Membrane Organization (GO:0061024) | 4 | cdc15, pep12, sec31, sec9 |
| Mitotic Cytokinesis (GO:0000281) | 4 | cdc15, imp2, pom1, sid2 |
| Protein-Containing Complex Assembly (GO:0065003) | 4 | bud6, tfs1, tif212, tif303 |
| Signaling (GO:0023052) | 3 | pom1, sck1, sid2 |
| Mitotic Cell Cycle Phase Transition (GO:0044772) | 2 | pom1, sid2 |
| Ribosome Biogenesis (GO:0042254) | 2 | lcp5, tif471 |
| Transcription, DNA-Templated (GO:0006351) | 2 | tfg1, tfs1 |
| Cell wall organization or biogenesis (GO:0071554) | 1 | scw1 |

Table EV3). Among these potential TORC1-dependent proteins, many were found to be conserved across eukaryotes, with 59% (49/83) sharing homologues in both budding yeast and humans, and a further 28% (23/83) with homologues identified in either one. Using experimental evidence annotated on the STRING database (Szklarczyk et al, 2019), we subsequently performed an interactome analysis on the 49 conserved proteins (see Materials and Methods) and found that 28 of these proteins were functionally or physically interacting. Within this group of interacting proteins, 4 main clusters emerged based on GO biological process categorisation: cytoplasmic translation, mRNA metabolism, vesicle-mediated transport and actin cytoskeletal organisation. This forms a core network of phosphoregulated proteins that may have roles in controlling cellular growth (Fig 3E; Table 2).

**The role of the S6 proteins kinases and translation initiation factor eIF4G on the control of protein synthesis rates**

This study has identified a range of putative targets that potentially regulate the rate of protein synthesis in response to TORC1. Of particular interest are the S6 proteins kinase (S6K) family, which are among the most widely discussed targets, hence we decided to further investigate and understand their potential role in TOR regulation of protein synthesis. There are 3 functional homologues of the S6Ks in fission yeast, Psk1, Sck1 and Sck2 (Nakashima 2012). After 40 min of Torin1 treatment, we found that the relative protein expression of all 3 proteins remained unchanged in both wild type

and tor1Δ cells (Fig 4A). Sck1 was identified in our interactome analysis as a potentially conserved TORC1-dependent target (Fig 3E), and one phosphosite in particular (T632) showed a greater than 2-fold phosphorylation decrease within 18 min of Torin1 addition (Fig 4B). Furthermore, in the earlier phosphoproteomics experiment, we also identified one Sck2 phosphosite that showed a more than 2-fold phosphorylation decrease in wild type cells within 30 min of Torin1 addition (Fig 4C). Previously identified downstream substrates for these protein kinases include the Rps601 and 602 proteins, which showed over 8-fold reductions within 40 min of Torin1 addition (Fig 2C and D). The changes in phosphorylation kinetics of the S6 protein kinases and their downstream targets therefore suggest that they could be potential modulators of protein synthesis downstream of the TOR kinase, as implicated in other studies (reviewed in (Saxton & Sabatini, 2017)).

To investigate this possibility, we undertook a genetic analysis of the S6K homologues. Previous genetic analyses have suggested that deletion of any one or combinations of the three S6K homologues does not affect cell viability in fission yeast (Nakashima et al, 2012). We therefore wanted to investigate whether the S6Ks have a role in modulating the rates of protein synthesis in response to growth inhibitory signals from the TOR pathway. We constructed a triple deletion mutant (S6K 3xΔ) of the S6 kinase homologues and measured the rates of protein synthesis following Torin1 treatment. After 60 min of Torin1 addition, we found no significant difference in the rates of inhibition in protein synthesis between S6K 3xΔ and wild type (Fig 4D). This result indicates that the immediate decrease in protein synthesis rates upon TOR inhibition is not dependent upon the S6Ks or their downstream targets.

Another set of putative targets of TOR-dependent regulation on protein synthesis are those related to cytoplasmic translation. These represent a notable proportion of the 27 proteins that are part of the conserved TORC1-dependent core interactome network, as identified in the cytoplasmic translation cluster (Fig 3E). In particular, 9 of the 27 proteins are encoded by genes essential for cell viability (outlined in white in Fig 3E), and are considered as stronger candidates for protein synthesis regulation given the central role protein synthesis has in cellular growth. Among these 9 proteins, 4 of them are Translation Initiation Factors (TIFs), and together with Lcp5, an essential ribonucleoprotein involved in 18S rRNA maturation, these 5 proteins play crucial roles in recruiting and engaging the ribosome for initiating translation. Additionally, one other translation-related protein Erf1 was identified in our analysis, which is shown to have functions at the end of the translational process, and is responsible for coordinating the release of newly synthesised peptides from the active ribosome.

Of these 6 essential translation-related proteins, 3 were found to have common phosphorylation sites in both our phosphoproteomics studies and showed greater than 2-fold changes within 40 min of Torin1 addition. Tif212 is the fission yeast homologue of the eukaryotic translation initiation factor eIF2 β-subunit and is one of the 3 subunits of the heterotrimeric GTP-binding protein required for the recruitment of the initiation methionine residue Met-tRNA$_i$. One Tif212 phosphorylation site was identified in the TORC1-dependency analysis, showing rapid dephosphorylation in both wild type and tor1Δ cells within 12 min of Torin1 addition (Fig 5A). Rapid phosphorylation changes were also observed for the translation release factor Erf1, with levels changing more than 2-fold in

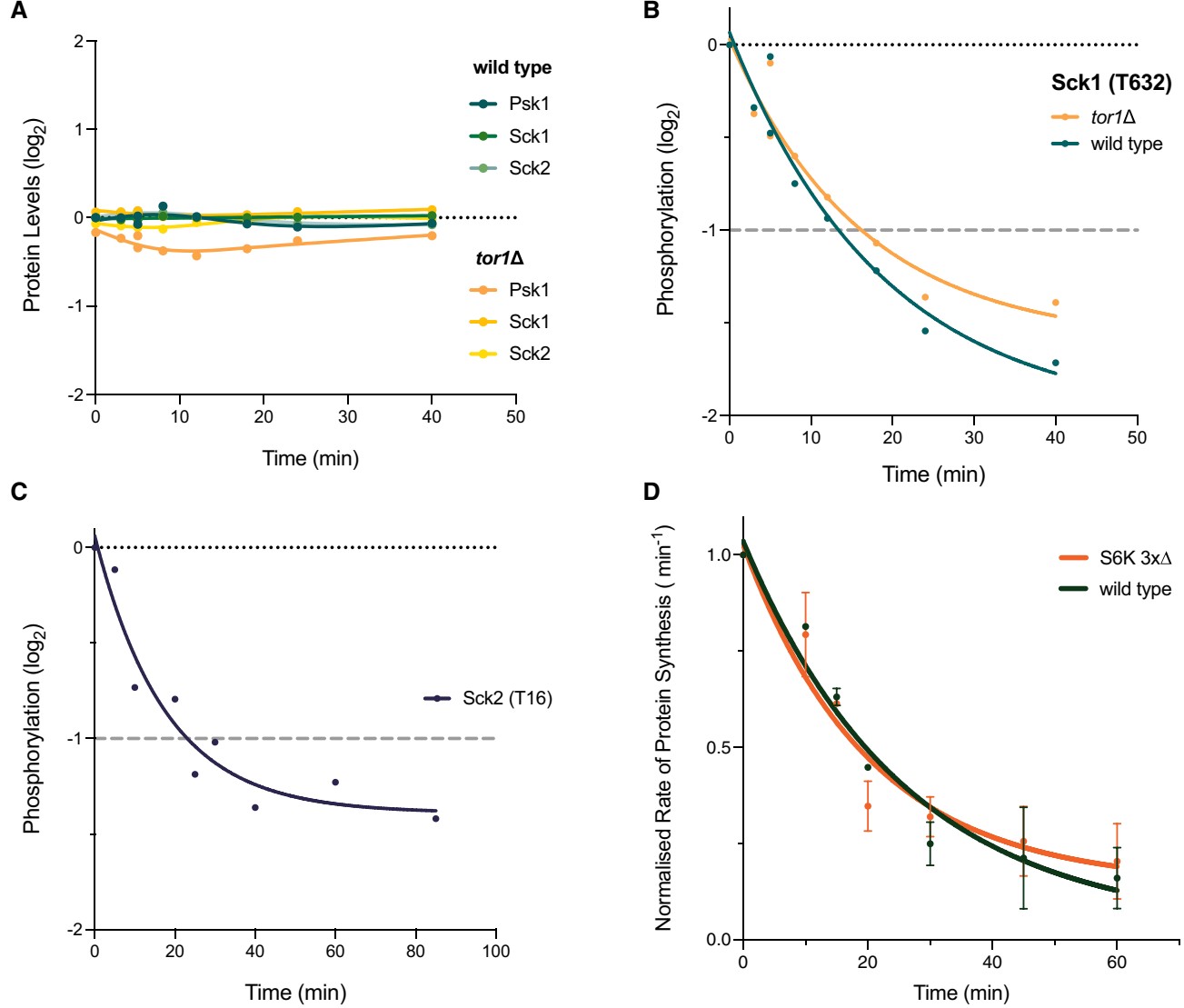

**Figure 4. The effects of TOR inhibition on S6K phosphorylation and rates of protein synthesis.**

Kinetic analyses on the identified phosphosites for the S6 kinase homologues in fission yeast, and the changes in rates of protein synthesis in the S6 kinase deletion mutant relative to wild type cells upon TOR inhibition.

A   Relative protein expression levels of the 3 S6 kinase homologues, Psk1, Sck1 and Sck2, plotted against time of Torin1 addition. Respective proteins are represented by the different coloured lines as detailed in the legend.

B   Relative phosphorylation level of the T632 phosphosite identified on Sck1 in the 40-min time course study. Phosphorylation levels of both wild type (green) and *tor1Δ* (yellow) are presented relative to the value for wild type cells at *T* = 0 min. Grey dashed line indicates 2-fold threshold.

C   Relative phosphorylation of the T16 phosphosite identified on Sck2 in the 85-min time course study (background adjusted detailed in Materials and Methods). Grey dashed line indicates 2-fold threshold.

D   Changes in the rates of protein synthesis upon Torin1 treatment (5 µM) measured for wild type and the S6 kinase deletion mutant (S6K 3xΔ) plotted over time. Non-linear regression exponential decay lines were fitted using the curve-fitting function on Prism9. Error bars represent the standard deviation (SD) of data aggregated from 3 independent experiments.

both wild type and *tor1Δ* cells within 10 min of Torin1 addition (Fig 5B).

Thirdly, the fission yeast homologue of eIF4G, Tif471 was identified as one of the most heavily phospho-regulated proteins, with a total of 27 phosphosites exhibiting TOR-dependent changes across the two studies. In particular, 10 sites showed rapid TORC1-dependent phosphorylation changes, all exceeding a 2-fold change by 30 min, and 6 showing a greater than 1.5-fold change within

10 min (Fig 5C and D). To determine whether the phosphosites identified on Tif471 play a regulatory role for protein synthesis, we constructed a non-phosphorylatable mutant of Tif471 and examined the rates of protein synthesis following Torin1 treatment. A total of 18 serine or threonine residues were replaced with alanine (see Materials and Methods for mutation sites), and the resulting *tif471_18A* mutant strain is viable. The mutant strain also has a similar growth rate to the control, which has a tagged wild type

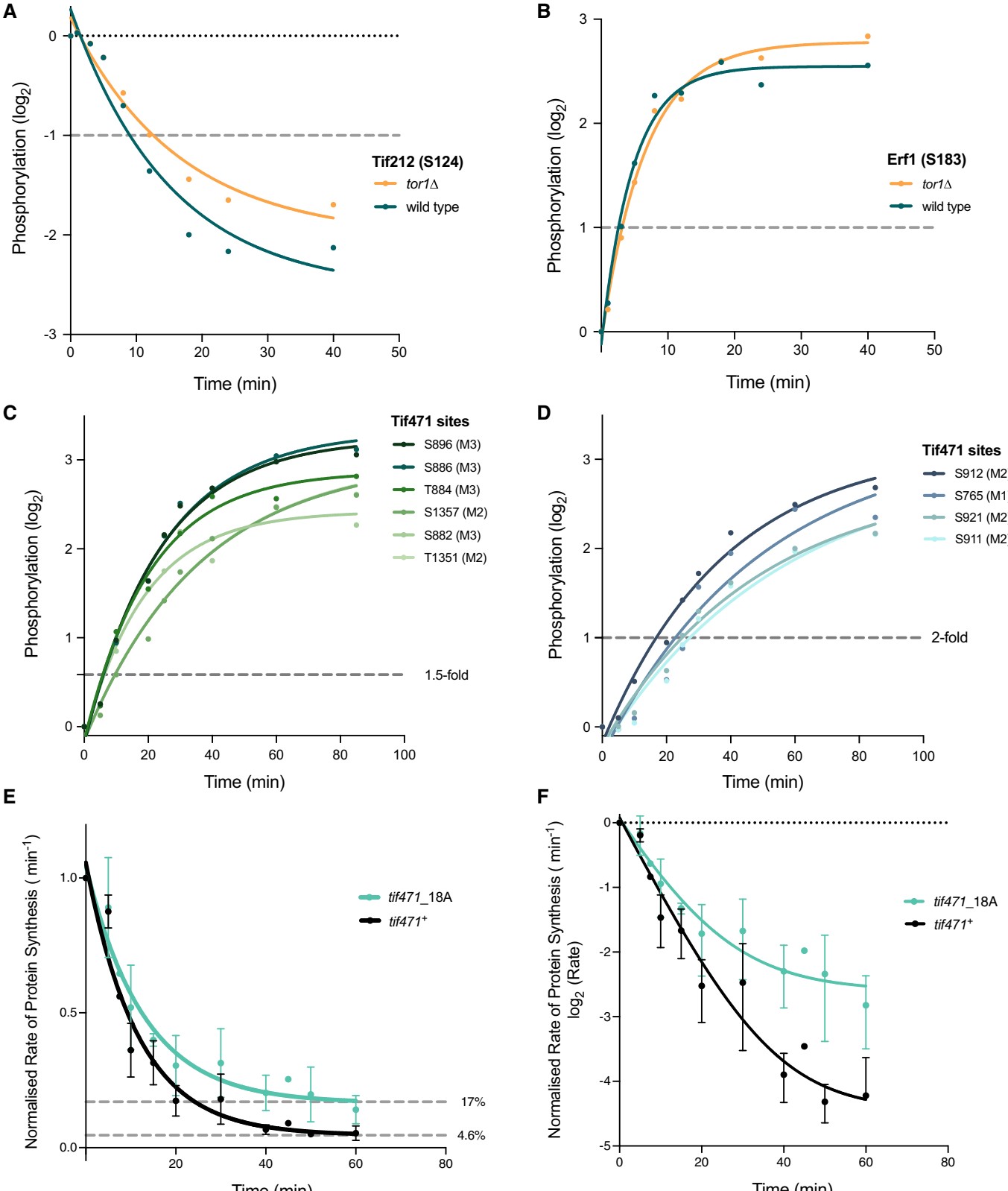

**Figure 5.**

◀

**Figure 5.  Examining potential essential translation-related targets and the functional importance of phosphorylation on rates of protein synthesis upon TOR kinase inhibition.**

Phosphorylation kinetics of common phosphosites identified for 3 essential translation-related proteins, and the effects of the *tif471*_18A phosphosite mutant on rates of protein synthesis upon Torin1 treatment.

A, B   Phosphosites identified on (A) Tif212 and (B) Erf1, respectively, showing 2-fold phosphorylation change within 20 min of Torin1 (5 μM) addition. Both wild type (green) and *tor1Δ* (yellow) phosphorylation values are relative to the starting phosphorylation levels before Torin1 addition in wild type cells ($T = 0$ min). Grey dashed lines indicate 2-fold threshold.

C, D   Relative phosphorylation levels of the 10 common phosphosites of Tif471 plotted against time after Torin1 treatment. Phosphorylation profiles are grouped based on phosphorylation kinetics, where 6 sites showed more than 1.5-fold increase by 10 min (C), and the remaining 4 exceeded 2-fold by 30 min (D). Grey dashed lines indicate 2-fold threshold.

E, F   Changes in rates of protein synthesis of the *tif471*_18A mutant compared to the wild type control (*tif471*[+]) upon Torin1 (5 μM) treatment. Residual rates of protein synthesis after 60 min of Torin1 treatment was 17% for the *tif471*_18A phosphomutant and 4.6% for the *tif471*[+] control relative to starting levels in the respective strains. The two graphs represent the untransformed (E) and $\log_2$ transformed (F) rates respectively. Error bars represent the standard deviation (SD) of data aggregated from three independent experiments.

sequence at the endogenous locus (*tif471*[+]) (Fig EV3), as well as similar protein expression levels (unpublished observations).

When comparing the rates of protein synthesis after 60 min of Torin1 treatment, we noted that the residual rate for *tif471*_18A was reduced to 17% of its starting level, over 3 times higher than the control (5%) (Fig 5E and F). This demonstrates that the changes in phosphorylation states of these Tif471 phosphosites contribute to the control of cellular rates of protein synthesis in response to TOR activity. As a reduction in protein synthesis rates still occurs in the *tif471*_18A mutant, other factors in addition to Tif471 will likely be involved in regulating protein synthesis in response to TOR inhibition, and are yet to be identified.

# Discussion

We have investigated how the TOR-signalling pathway regulates eukaryotic cell growth, with a focus on understanding the control of global cellular protein synthesis. Using fission yeast, we describe an in-depth analysis of the proteomic and phosphoproteomic changes that occur after inhibition of both TOR protein kinases with Torin1 in two time course studies at fine temporal resolution. Our results establish that TORC2, which contains the non-essential Tor1 kinase, is dispensable for regulating global cellular protein synthesis, and so we have characterised changes that are primarily dependent on the essential Tor2 kinase containing TORC1. A total of 20,894 phosphosites located on 2,306 different phosphoproteins were identified in the two studies. Of these, 908 phosphosites on 343 proteins exhibited a more than 2-fold change in phosphorylation levels after 40 min of Torin1 treatment, by which time protein synthesis rates were reduced over 90%. Two-thirds of these proteins (213/316) shared homologues in both budding yeast and human cells, and a total of 88% (279/316) of these had homologues in at least one of these two organisms.

Our database of phosphoproteomic changes after TOR kinase inhibition provides an in-depth resource for investigating the TOR pathways' involvement in regulating cell growth, with many of the identified phosphoproteins conserved among eukaryotes. We investigated two possible candidates, the S6 protein kinases and the translation initiation factor, eIF4G (Tif471), examining their potential roles in cell growth control with respect to protein synthesis. The S6 kinases have widely been suggested as downstream regulators of the TOR kinase pathways and implicated in growth control via modulation of translation initiation activity (see reviews in

(Wullschleger *et al*, 2006; Ma & Blenis, 2009; Saxton & Sabatini, 2017)). However, following deletion of all three S6 kinase fission yeast homologues (Psk1, Sck1 & Sck2), we observed no effects on the rate of rapid protein synthesis reduction upon inhibition of the TOR kinases.

One of the proteins most heavily regulated by phosphorylation identified in our study was the translation initiation factor Tif471, the fission yeast homologue of eIF4G. In a phosphomutant of Tif471 (*tif471*_18A), we find that in response to TOR inhibition, the residual rate of protein synthesis is reduced to 17% of the starting level, more than three times higher than in wild type. This, to our knowledge, is the first demonstration of a component regulated by phosphorylation as a consequence of TOR inhibition that has an immediate impact on reducing the global cellular rates of protein synthesis. The partial inhibition observed suggests that there are multiple points of control distributed within the TOR downstream network, and complete inhibition will require the coordination of various components.

The detailed temporal and molecular resolution of our study provides a comprehensive data resource for further evaluation of potential TOR-dependent regulatory targets, and the accompanying data visualisation tool allows for the data to be readily accessible. Gene ontology (GO) analyses of the TOR-dependent phosphoproteome identified a range of molecular processes with four broad categories that were particularly enriched: signalling-related processes, cell polarity and actin cytoskeletal organisation, vesicle-mediated transport and autophagy, and translation-related processes.

The switching off of growth will require a coordinated response from multiple major cellular processes, so it is unsurprising that several signalling network pathways have been implicated in TOR regulatory control. The MAPK-signalling pathway is thought to be responsible for maintaining structural integrity of a cell through actin cytoskeletal reorganisation (Samaj *et al*, 2004; Madrid *et al*, 2016). Our study identified the MAPKK homologue Wis1, and downstream signalling cascade components including the RAS protein homologue Ras1, and regulators of the Rho GTPases, Rga1 and Rgf1. In addition, components of the SIN-MOR pathways, Sid2 and Nak1, responsible for coordinating cytoskeletal rearrangements in fission yeast (Gupta *et al*, 2013) were also identified. Protein homologues of these pathways are conserved in mammalian cells as part of the Hippo and NDR-kinase signalling networks and have been implicated in growth control. These proteins are also often found to be misregulated in cancer (Halder & Johnson, 2011).

Across the eukaryotes, components of the actin cytoskeletal network and those involved in cellular spatial organisation have previously been implicated as targets of the non-essential Tor1 containing complex TORC2 (Loewith *et al*, 2002; Jacinto *et al*, 2004; Sarbassov *et al*, 2004; Cybulski & Hall, 2009; Rispal *et al*, 2015). In particular, TORC2 has been identified as a key component of PI3K/ Akt signalling (Sarbassov *et al*, 2004; Hresko & Mueckler, 2005), and in fission yeast, it has been suggested that it is regulated by the Rab-family GTPase, Rhy1 and its regulators (Tatebe & Shiozaki, 2010; Tatebe *et al*, 2010). In this study, we identified a conserved component of the GEF complex Sat1, which is involved in Rab GTPase activation to regulate TORC2-Akt signalling (Tatebe *et al*, 2010). Interestingly, phosphosites on proteins involved in actin cytoskeletal organisation including Bud6 and Imp2 also exhibited TORC1-dependent phosphorylation changes upon addition of Torin1, suggesting a potential overlap between the TORC1 and TORC2 complexes in the remodelling aspects of cytoskeletal structures within a cell.

In line with a recent fission yeast study, where vesicle-mediated transport proteins were altered by TOR-dependent cellular nutritional states (Lie *et al*, 2018), we showed that phosphorylation of vesicle-mediated transport and autophagy-related proteins are dependent on TOR activity. A number of essential cellular growth functions are supported by vesicular transport processes, including Golgi trafficking, exocytosis and endocytosis, as well as proteosomal and lysosomal pathways. Therefore, coordination of anabolic and catabolic processes appears to be involved in cellular decision-making required for cell growth, perhaps in part regulated through interplay between TOR and autophagy-related processes (reviewed in (Dunlop & Tee, 2014)).

Consistent with previous studies suggesting translation initiation as a major point of control for a TOR-dependent regulation of growth (reviewed in (Ma & Blenis, 2009; Loewith & Hall, 2011)), a number of the translation initiation components were identified in our study. Notably, 5 of the 7 translation-related proteins categorised as part of the TORC1-dependent core interactome network were translation initiation factors (Tif-). Other components of the translation machineries were also identified, including Erf1, the translation termination and release factor, and Lcp5, the U3 snoRNP-associated protein, shown to be involved in 18S rRNA maturation in budding yeast (Wiederkehr *et al*, 1998). The human orthologue of Lcp5, Neuroguidin (NGDN) has also been shown to repress translation through interactions with the cap-binding eukaryotic initiation factor 4E (eIF4E), preventing the eIF4E-4G interaction require for mRNA cap-dependent translation initiation (Jung *et al*, 2006). This supports the potential role of eIF4G in translational control.

In addition, our study revealed a cluster of conserved proteins related to mRNA metabolism that exhibited TORC1-dependent phosphorylation changes, and some of these proteins have been shown to interact with translation initiation machineries. These include Ccr4 and Not2, which are components of the CCR4-Not complex, and have been shown to regulate mRNA fate and turnover (Parker & Song, 2004; Wiederhold & Passmore, 2010). Ccr4 co-localises with P-bodies and is essential for P-body formation in mammalian cells (Cougot *et al*, 2004; Andrei *et al*, 2005). The P-body exoribonuclease Sts5, which is the homologue of SSD1 in budding yeast, was also identified in our study. SSD1 is shown to colocalise with mRNAs associated with growth zones, and was implicated in P-body and stress-granule dependent translational repression, after exposure to cellular stress in budding yeast (Jansen *et al*, 2009; Kurischko *et al*, 2011). This suggests that TOR-dependent translational regulation is not limited to the regulation of translation initiation machineries, and may also involve factors that influence mRNA fate and turnover.

We have demonstrated that protein synthesis rates are rapidly reduced upon TOR-dependent growth inhibition and that this reduction is not dependent on S6 kinase activity. We propose that translation-related proteins remain key potential candidates for regulating eukaryotic cell growth, and identify the eIF4G homologue in fission yeast, Tif471, as one of the regulators of protein synthesis inhibition through TOR-dependent phosphorylation. Our list of potential TOR phosphorylation targets shows that TOR-dependent growth control requires coordination of a range of cellular processes for timely and effective regulation, and our study provides a foundational resource for potential targets to be further investigated in both fission yeast and other eukaryotes.

# Materials and Methods

## Reagents and Tools table

| Reagent/Resource | Reference or source | Identifier or catalog number |
|---|---|---|
| **Experimental models** | | |
| *S. pombe: 972 h-* (wild type) | Nurse Lab | PN1 (TM11) |
| *S. pombe: tor2_G2040D (Torin1-resistant)* | Petersen Lab (Atkin *et al*, 2014) | JP1669 (TM134) |
| *S. pombe: tor1Δ::HphMX6 h-* | This study | TM97 |
| *S. pombe: psk1Δ::HphMX6 Sck1Δ::NatMX6 Sck2Δ::KanMX6 h-* (S6K 3xΔ) | This study | TM74 |
| *S. pombe: 972/972h-/h-* | This study | TM21 |
| *S. pombe: tif471+-v5-HphMX6 h-* | This study | TM121 |
| *S. pombe: tif471_18A-v5-HphMX6 h-* | This study | TM156 |

**Reagents and Tools table**  (continued)

| Reagent/Resource | Reference or source | Identifier or catalog number |
|---|---|---|
| **Recombinant DNA** | | |
| Plasmid: pSC-A-Amp-tif471(wt)-v5-HphMX6 | This study | TMp16 |
| Plasmid: pSC-A-Amp-tif471_13A-v5-HphMX6 | This study | TMp20 |
| Plasmid: pSC-A-Amp-tif471_18A-v5-HphMX6 | This study | TMp21 |
| **Oligonucleotides and sequence-based reagents** | | |
| PCR primer: ATTCCTGAAGGATCTGATGGAAGC | This study | pTM101 |
| PCR primer: CATTAGACTTAAAGGAGCCGCA | This study | pTM102 |
| PCR primer: TTTGGAAAATTTTAATTTCATTC | This study | pTM111 |
| PCR primer: TTTACCCTCTTCCTCAGC | This study | pTM112 |
| PCR primer: acggaaagctgaggaagagggtaaaCGGGAAGCTGATAAAAAC | This study | pTM113 |
| PCR primer: atgaatgaaattaaaattttccaaaGCATTTCTCAATTACAATGTAG | This study | pTM114 |
| PCR primer: gaatcgatcgaggtttcgccgcttctggtgctggat | This study | pTM141 |
| PCR primer: atccagcaccagaagcggcgaaacctcgatcgattc | This study | pTM142 |
| PCR primer: ctggttttggtggtcctgcagagagaaagggcatt | This study | pTM143 |
| PCR primer: aatgccctttctctctgcaggaccaccaaaaccag | This study | pTM144 |
| PCR primer: ctctcgatccggcgcaaatgctcacgcccatgctggccct | This study | pTM145 |
| PCR primer: gggccagcatgggcgtgagcatttgcgccggatcgagag | This study | pTM146 |
| PCR primer: ACCGACTTTTCTGCTTTGGT | This study | spTM64 |
| PCR primer: AGCAATCCACCATTTAGCTCT | This study | spTM67 |
| PCR primer: TCCGTGATGACTTACACCGC | This study | spTM72 |
| PCR primer: TGAGTTAGAATCTTTTTGACCCCC | This study | spTM75 |
| **Chemicals, enzymes and other reagents** | | |
| Complete mini Protease Inhibitor Cocktail | Sigma Aldrich | Cat # 11836153001 |
| PhosSTOP phosphatase Inhibitor tablets | Sigma Aldrich | Cat # PHOSS-RO |
| Propidium iodide solution | Biotium | Cat # 40017 |
| Torin1 | Torcris | Cat # 4247 |
| Pierce Trypsin Protease, MS Grade | ThermoFisher | Cat # 90058 |
| **Software** | | |
| FlowJo v10.3.0 | FlowJo | https://www.flowjo.com/ |
| Prism v9.0.0 | GraphPad | https://www.graphpad.com/scientific-software/prism/ |
| Rstudio v1.3 | Rstudio | https://rstudio.com/products/rstudio/rstudio-desktop |
| Perseus v1.4.0.2.3 | Perseus | https://maxquant.net/perseus/ |
| MaxQuant v1.5.0.13 | MaxQuant | https://www.maxquant.org/ |
| **Online Databases** | | |
| PomBase | PomBase | https://www.pombase.org/ |
| PANTHER (GO Enrichment) | Gene Ontology | http://geneontology.org/ |
| Clustal Omega | EMBL-EBI | https://www.ebi.ac.uk/Tools/msa/clustalo/ |
| **Deposited Data** | | |
| Mass spectrometry files | This study | https://www.ebi.ac.uk/pride/archive/ PRIDE: PXD023664 |
| Data Visualisation Tool | This study | https://tmak.shinyapps.io/TM_App-Table_EV3/ |
| **Other** | | |
| Ultrospec 2100 pro UV/visible spectrophotometer | Amersham Biosciences | Cat # 80-2112-21 (BioChrom) |
| Click-iT® HPG Alexa Fluor® Protein Synthesis Assay Kits | ThermoFisher | Cat # C10428 |
| Dynabeads Protein A | ThermoFisher | Cat # 10002D |

**Reagents and Tools table**   (continued)

| Reagent/Resource | Reference or source | Identifier or catalog number |
|---|---|---|
| Dynabeads M-270 Epoxy | ThermoFisher | Cat # 14302D |
| TMT 10plex Isobaric Label Reagent Set 1 × 0.8 mg | ThermoFisher | Cat # 90110 |
| High-Select™ TiO2 Phosphopeptide Enrichment Kit | ThermoFisher | Cat # A32993 |
| High-Select™ Fe-NTA Phosphopeptide Enrichment Kit | ThermoFisher | Cat # A32992 |
| Pierce™ High pH Reversed-Phase Peptide Fractionation Kit | ThermoFisher | Cat # 84868 |
| EASY-Spray C18 column, 75 mm × 50 cm | ThermoFisher | Cat # ES803 |
| Orbitrap Fusion Lumos Tribrid Mass Spectrometer | ThermoFisher | Cat # IQLAAEGAAPFADBMBCX |
| BD LSRFortessa | BD Biosciences | Cat # 649225 |
| UltiMate 3000 HPLC System | ThermoFisher | Cat # 5041.0010 |
| QuikChange Multi Site-Directed Mutagenesis | Agilent | Cat # 200514 |

## Methods and Protocols

### S. pombe genetics and cloning

Full genotype of all strains, plasmids, gDNA fragments and primers used in this study are listed in the Reagents Table. Standard *S. pombe* media and methods as previously detailed in (Moreno *et al*, 1991). Gene deletions and tagging fragments were prepared using PCR based amplifications described in (Bahler *et al*, 1998), with primers designed on the Bähler laboratory website (Resources). S6K 3xΔ mutant was generated through consecutive lithium acetate transformations and selection, and verified through colony PCR checking of both the gene deletion construct and absence of wild type gene copies.

The *tif471*_18A mutant was constructed by first tagging the *tif471* gene at the endogenous locus with a v5-HphMX tag at the 3′ end of the open reading frame (ORF) (subsequently used as the *tif471*+ control). The *tif471* genes were amplified from the extracted by PCR with primers (pTM101 & 102) 1.06 kb either side of the ORF. The amplified gene was subcloned into the pSC-A-Amp vector using the StrataClone Blunt PCR Cloning Kit (TMp16). The 18 alanine mutations were introduced through 2 consecutive cloning steps, first replacing amino acid position 706 onwards until the end of the ORF with the synthetic gene fragment (gBlock) synthesised by Integrated DNA Technologies (IDT). The gBlock fragment contained 13 alanine mutations: S912A, S919A, S921A, S1293A, S1300A, S1333A, S1349A, T1351A, S1353A and S1357A. The 3′ end the endogenous gene on TMp16 was replaced by the fragment at the using the NEBuilder HiFi DNA Assembly kit and primers pTM111-114. The final 5 alanine mutations, S882A, T884A, S886A, S896A and S896A, were added by 2 subsequent reactions using the Agilent QuikChange Multi Site-Directed Mutagenesis Kit and primers pTM141-146, resulting in the TMp21 plasmid containing all 18 alanine mutants on Tif471.

The entire locus containing mutated amino acid residues was then amplified by PCR with pTM101 and 102 and purified using the QIAquick PCR Purification Kit, then transformed into stable wild type diploid cells (TM21) and selected with antibiotic resistance. The resultant heterozygous diploid was subsequently sporulated into haploids, to check for a 50:50 segregation of wild type to mutant haploids through random spore analysis, and mutant haploids were isolated. Alanine mutations were checked with specifically designed primers on Primer3Plus website (spTM64, 67, 72 and 75) by DNA sequencing from extracted gDNA.

### Growth rate measurement and calculations

Cells were inoculated in static liquid media overnight at corresponding temperature and subsequently diluted to within exponential growth cell density (OD ~ 0.1 - 0.8). Optical density (OD) was measured at wavelength ($\lambda$) = 595 nm using the Ultrospec 2100 pro UV/visible spectrophotometer (Amersham Biosciences). $OD_{595} = 0.1$ approximated to $1.26 \times 10^6$ cells/ml. All cell cultures were maintained in exponential growth for 48 h in the culturing conditions defined above before any experiments were conducted. Growth rate calculations were made based on the OD measurements, where the logarithm (base 10) of the OD was taken (Log (OD)) and plotted against time. Using the linear regression line fitting function on Prism9, the gradient of the fitted line was used to calculate the generation time (*gt*) using the formula $gt = \frac{\log_{10}(2)}{m}$, where *m* is represented by the gradient of the regression line. Error bars indicate standard deviations (SD) of three independent experiments unless stated otherwise.

### Protein synthesis assay

Cells were grown in EMMG at 25°C and maintained in exponential growth (OD 0.1–0.8) for 48 h in liquid culture; cells were diluted to OD 0.4–0.5 for the start of the assay. The main culture was then split into individual cultures for each time point and allowed to recover for at least 1 h before beginning the assay. The individual cultures were treated with Torin1 (added at 1:1,000 from 5 mM stock in DMSO to final concentration 5 μM) for different lengths of time accordingly. The methionine analogue L-homopropargylglycine (HPG) was added (from a 50 mM stock solution to a final concentration of 50 μM) to cultures 15 min before the first sampling of each time point. Two sample were taken at 10 min apart for the individual time points, and the rate of protein synthesis was calculated from the difference between the two values and then divided through by 10 min.

### Sample processing

420 μl of samples were collected at the indicated time points and fixed in 980 μl of 100% ice-cold ethanol to reach final concentration of 70% ethanol and then kept on ice for 20–30 min. Sample processing and labelling procedures were based on (Knutsen *et al*, 2015). Cells were washed and permeabilised for 20 min with 0.5ml of PBS and 1% Triton X-100 in PBS, respectively, then washed and blocked in 0.5 ml of 1% BSA in PBS for 1 h. All steps were performed at

room temperature and incubations were with rotation on tube rotators, protected from light.

Composition of reaction mix was modified from the Click-iT™ HPG Alexa Fluor™ 488 Protein Synthesis Assay Kit (Invitrogen). Each sample was stained with 375 µl of reaction mix per reaction, containing 321.56 µl of 1× Click-iT HPG reaction buffer, 15 µl Copper (II) sulphate ($Cu_2SO_4$) (100 mM), 0.94 µl 488 Alexa Fluor® azide and 37.5 µl 1× Click-iT HPG buffer additive. Reaction mix was removed, and samples were washed with 500 µl of sodium citrate (50 mM). Cells were subsequently washed and resuspended in 500 µl of sodium citrate and treated with RNase (0.1 mg/ml) for 3 h to overnight at 37°C, protected from light. Following RNase digestion, cells were pelleted and resuspended in 500 µl of sodium citrate with propidium iodide (PI) added to a final concentration of 2 µg/ml. Samples were loaded onto 96-well U bottom plates and ran on BD-LSR-Fortessa flow cytometers using the High Throughput Sampler (HTS), and 100,000 events were recorded per sample. Sample flow rate was set to 1 µl/s, with a mixing volume of 100 µl for 2 mixes at 180 µl/s mixing speed, and a wash volume of 400 µl.

Voltage for corresponding parameters was set based on defined criteria. Side scatter (SSC) adjusted to have ~90% of the cells between 7500 and 15,000 (roughly one doubling). Voltage for the 530/30 blue laser channel (for detection of 488) was set to have the maximum range of intensity whilst keeping 95% of the events below the maximum 250,000 (AU). For the 610/20 yellow laser channel (PI), voltage was set to have the 2C DNA content peak centred around 50 (AU). Cells were manually gated on FlowJo (version.10.3.0) for doublet discrimination, and only cells with 2C DNA content or below were included for further analysis. For the initial experiment in establishing HPG incorporation as a measure for protein synthesis at steady state, median values were calculated for whole populations after data clean-up and values were exported for statistical analysis on R. Linear regression lines were fitted using the linear model of the geom_smooth function. Error bars indicate standard deviations (SD) of three independent experiments unless stated otherwise.

### Proteomic and phosphoproteomic studies

Two time course studies were conducted to investigate the proteomic and phosphoproteomic changes upon Torin1 (5 µM) addition, with the first time point ($T = 0$ min) taken just before Torin1 or DMSO addition. A DMSO sample treated for the duration of the respective time courses was used as a control to account for non-specific phosphorylation changes. Time points for the 85-min time course were taken at 0, 5, 10, 20, 25, 30, 40, 60 and 85 (+DMSO 85) min, and for the 40-min time course, they were at 0, 1, 3, 5, 8, 12, 18, 24 and 40 (+DMSO 40) min.

### Sample preparation for mass spectrometry

Each protein sample (300 µg) was reduced with 5 mM dithiothreitol (DTT) for 25 min at 56°C, alkylated with 10 mM iodoacetamide for 30 min at room temperature in the dark and then quenched with 7.5 M DTT. Samples were digested using SP3 on-bead methodology (Hughes *et al*, 2019) with the variation that 50 mM HEPES (pH 8.5) was used in place of ammonium bicarbonate. Briefly, proteins were bound to the SP3 beads (10:1 beads:protein (w/w)) in 50% ethanol

(v/v) and then washed three times in 80% ethanol, prior to resuspension in 50 mM HEPES (pH 8.5) with 1:40 (trypsin:protein (w/w)) overnight at 37°C. The digested samples were arranged in sets of ten and labelled using the TMT10plex Isobaric Label Reagent Set (Thermo Fisher), as per the manufacturer's instructions. Following labelling and mixing, multiplexed samples were desalted using a C18 SepPak column. Phosphopeptide enrichment was performed by Sequential Enrichment from Metal Oxide Affinity Chromatography (SMOAC, Thermo Fisher) with initial enrichment using the High-Select TiO2 Phosphopeptide Enrichment Kit followed by the High-Select Fe-NTA Phosphopeptide Enrichment Kit (both Thermo Scientific) for the non-bound flow through fractions. Phosphopeptides and non-bound flow through fractions were desalted and fractionated using the High pH Reversed-Phase Peptide Fractionation Kit (Pierce) and analysed on an Orbitrap Fusion Lumos mass spectrometer (Thermo Fisher) coupled to an UltiMate 3000 HPLC system for online liquid chromatographic separation. Each run consisted of a 3 h gradient elution from a 75 µm × 50 cm C18 column.

MaxQuant (version 1.5.0.13) was used for all data processing. The data were searched against a PomBase (Lock *et al*, 2019) extracted *S. pombe* proteome FASTA file. Default MaxQuant parameters were used with the following adjustments with Phospho(STY) being added as a variable modification, MaxQuant output files were imported into Perseus (version 1.4.0.2).

### Quantification and statistical analysis

All reporter intensities were $\log_2$ transformed and only phosphosites or proteins which were quantified in all 20 channels were retained. Reported intensities were normalised by subtracting the median value for a specific TMT channel and then by subtracting the median value of each phosphosite or protein across the time course. Both protein and phosphorylation levels were compared to the corresponding values measured in wild type cells at 0 min before Torin1 or DMSO (control) addition for the two respective experiments.

### Proteomic analysis

Changes in protein levels are normalised to the starting protein levels at time $T = 0$ min in wild type cells for the respective experiments before Torin1 or DMSO addition (labelled "wt.0.min" and "WT 00 min" for the 85-min and 40-min experiments, respectively, in Table EV1). For both time course studies, proteins showing non-specific (DMSO dependent) changes were excluded from subsequent analysis if they showed more than 1.5-fold change (± 0.585) in the DMSO treated sample at the end of the respective time courses, either at 85 ("wt.85.DMSO") or 40 min ("WT 40 min DMSO"). For the 85-min time course experiment, proteins were excluded in further analysis if they showed more than 1.5-fold difference (within the ± 0.0585 range) in starting protein levels in the Torin1-resistance strain ("tr.0.min" in Table EV1). Frequency distribution analysis was carried out on values from the "wt.40.min" column (Table EV1) following data clean-up in Prism9 ($n = 3,380$, bin width = 0.2). For the 40-min time course study, proteins that showed more than a 1.5-fold difference (± 0.585) between the 40-min Torin1-treated sample and the DMSO control in either wild type ("WT 40 min - DMSO control") or *tor1Δ* ("Tor1D 40min - DMSO control") cells were excluded from subsequent analysis in order to account for DMSO non-specific changes. (Columns mentioned highlighted in green in Table EV1).

### Phosphoproteomic analysis

Phosphosites were first filtered for more than 2-fold change in phosphorylation in wild type cells ("wt.85.min" and "WT 40 min") at the end of the respective 85-min and 40-min time course experiments (Tables EV2 and EV3). Non-specific DMSO changes were accounted for by only including sites that showed more than 2-fold difference in phosphorylation after subtracting values of the DMSO treated control from the Torin1-treated samples, respectively, i.e. "wt 85min - DMSO control" column for the 85-min experiment, and the "WT 40min - DMSO control" column for the 40-min experiment. For the 85-min time course experiment, sites that showed more than 1.5-fold ($\pm$ 0.585) change in phosphorylation levels at the end of the time course ($T = 85$ min) relative to the start ($T = 0$ min) in the Torin1-resistant strain ("tr.85.min relative control") were excluded. Phosphorylation levels were subsequently corrected for non-specific Torin1-related changes in the 85-min time course by subtracting the phosphorylation values of the Torin1-resistant strain from the values measured in wild type cells in the corresponding time points for the individual phosphosites. Values are then background subtracted to have the relative phosphorylation levels before Torin1 addition ($T = 0$ min) starting at zero for direct comparison among sites and subsequent fold change analyses (labelled "sub.x.min" in Tables EV2 and EV3; header in blue). Sites exhibiting more than 2-fold change by 40 min of Torin1 treatment ("sub.40.min") and sustaining the change until the end of 85 min ("sub.85.min") were considered potential TOR-dependent phosphosites.

In the 40-min time course, potential sites were further categorised as TORC1- or TORC2-dependent based on the phosphorylation changes in wild type and *tor1Δ* cells relative to their respective phosphorylation levels before Torin1 addition ($T = 0$ min). Sites showing more than 2-fold change by 40 min in wild type cells ("WT 40 min" and "WT 40min - DMSO control") but less than 1.3-fold in *tor1Δ* ("Tor1D norm 40 min" and "Tor1D 40min - DMSO control") were classified as TORC2-dependent sites, and TORC1-dependent sites were defined as those showing more than 2-fold change in both wild type and *tor1Δ* cells (marked on Table EV3 respectively as "TORC2" and "TORC1" under "Complex dependency"; one of the filtering criteria on the online Data Visualisation Tool). The change in relative phosphorylation levels is represented in XY plots where the x-axis indicates time after Torin1 addition and the y-axis shows the level of phosphorylation relative to the starting level before Torin1 addition. All plots for the 85-min time course represent the background corrected and zero normalised values ("sub.x.min"; x indicates the corresponding time points, corresponding column titles highlighted in blue in Table EV2) unless stated otherwise. Plots for the 40-min time course represent phosphorylation values that are both normalised to the started values at wild type $T = 0$ min ("WT XX min" & "Tor1D norm XX min"; XX indicates the corresponding time points, column titles highlighted in blue in Table EV2). Lines represented on the graphs are drawn using the curve-fitting function on Prism9 to fit one-phase exponential decay non-linear regressions unless stated otherwise.

### Data visualisation tool

A data visualisation tool was developed for visualising the phosphorylation data of the 908 TOR-dependent phosphosites from the two phosphoproteomics time courses (data based on Table EV3). An illustrated example of the application interface is provided in Fig EV1A–D, and a publicly accessible link is made available online on https://tmak.shinyapps.io/TM_App-Table_EV3/. Tabs on the top bar provide individual access to the "85-min study" and "40-min study" datasets, respectively, with subsequent filtering selections specific to each dataset (Fig EV1A). Prompts on the graph place-holders in grey specify the filtering selections required for plotting and displaying the graphs.

For the "85-min study" tab, the "Conserved in" selection allows for sub-setting of the protein list based on conservation, and subsequently, the list of proteins available for selection gets updated in the "Protein" field based on the selected criteria (Fig EV1B). A graph in the top panel is then displayed with all the phosphosites of the selected protein plotted, with the corrected phosphorylation values plotted over time in minutes. Further information on the individual sites including specific phosphorylation values can be accessed by hovering the mouse over each line at the specific time point of interest (Fig EV1B). Individual phosphosites can also be specifically selected or unselected by single-clicking on the site of interest in the legend on the right (Fig EV1C). Double-clicking on an individual site specifically selects and displays that site in isolation. The "Site" field in the sidebar panel is updated according to the selected protein, and multiple sites can be selected for visualisation of the individual phosphosites in the lower facet panels (Fig EV1C). The "Strain" field provides an option for the data from the independent strains or the corrected relative phosphorylation values (selected by default) to be displayed in isolation or overlaid onto the same graph for each individual phosphosite in the lower panel. All lines were fitted with a non-linear regression using the "nls" method in the stat_smooth Self-Starting Nis Asymptotic Regression Model function from the R ggplot package. Note that if a line cannot be fitted due to a "singular gradient" error, the line will be omitted and only points are plotted on the graph.

In the "40-min study" tab, similar navigation and selection fields can be found, but the "TOR Complex dependency" filter is in place of "Conserved in" field in the other panel. A further sub-categorisation selection field is included between the "Protein" and "Site" fields to select for sites that may or may not be in common with the 85-min dataset (Fig EV1D). The "Strain" field allows for the phosphorylation profiles of wild type (selected by default) or *tor1Δ* strains to be displayed individually or overlaid on the same graph for each phosphosite. The "Wrap" field provides the option of displaying the facet panels in the specified number of rows selected from the dropdown menu and is available on both tabs. Both the displayed plots can also be downloaded in either of the tabs by using the "Download Plot (Sites)" button in the sidebar panel for downloading the faceted plots of the individual phosphosites, or the camera icon as shown in Figure EV1D for the graph in the top panel.

### GO categorisation and enrichment analysis

The corresponding proteins of the identified phosphosites were categorised based on Gene Ontology (GO) biological processes (Tables 1 and 2) as annotated on PomBase (Lock *et al*, 2019). Note that proteins can belong to more than one corresponding category. Gene deletion viability and information on protein conservation and homologues were also based on PomBase annotations. Subsequent GO enrichment analysis was carried out for the 149 conserved proteins from the 85-min time course that contained annotated

homologues in both *Saccharomyces cerevisiae* and human. The analysis was performed using the enrichment analysis tool from the PANTHER classification system (Mi *et al*, 2013) with respect to the GO biological process in *S. pombe*, and the provided reference list was based on the list of identified proteins from the proteomic analysis from the same study. Table 2 is arranged and ranked according to the fold enrichment and the categories are representative of processes that showed more than a 2-fold enrichment.

### Interactome analysis

The analysis was performed on the 49 conserved proteins from the 83 proteins that were identified to contain TORC1-dependent phosphites. Using the STRING database (Szklarczyk *et al*, 2019), the potential interactions were scored based on both functional and physical interactions from experimental, text mining and database sources only. Two nodes were connected upon reaching a minimum score of 0.4, and the lines were weighted to indicate the relative strength of the annotated interactions. 28 proteins are represented on the interactome network and are marked on Table EV3.

## Data availability

The datasets produced in this study are available in the following databases: Mass spectrometry proteomics data: PRIDE PXD023664 (https://www.ebi.ac.uk/pride/archive/projects/PXD023664).

Expanded View for this article is available online.

## Acknowledgements

This work was supported by the Francis Crick Institute which receives its core funding from Cancer Research UK (FC01121), the UK Medical Research Council (FC01121) and the Wellcome Trust (FC01121). In addition, this work was supported by the Wellcome Trust Grant to PN [grant number 214183 and 093917], The Lord Leonard and Lady Estelle Wolfson Foundation and the Woosnam Foundation and Breast Cancer Research Foundation (BCRF-20-117). For the purpose of Open Access, the author has applied a CC BY public copyright licence to any Author Accepted Manuscript version arising from this submission. We are grateful to Scott Curran, Jessica Greenwood, Jacky Hayles, Clovis Basier, Emma Roberts and Ryoko Mandeville for critical reading of the manuscript and help with experiments. The Protein Analysis and Proteomics Platform at the Crick for mass spectrometry, Hefin Rhys for critical advice on data visualisation tool development and Janni Petersen for strains.

## Author contributions

TM and PN conceived the experiments. TM designed and performed the experiments. AWJ performed mass spectrometry and processed raw data. TM analysed processed data. TM and PN wrote the manuscript.

## Conflict of interest

The authors declare that they have no conflict of interest.

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
