## [Review Process File · The EMBO Journal]

The TOR-dependent phosphoproteome and regulation of cellular protein synthesis

Tiffany Mak, Andrew Jones, and Paul Nurse

DOI: [10.15252/embj.2021107911](https://doi.org/10.15252/embj.2021107911)

Corresponding author(s): Tiffany Mak (tifmak@biosustain.dtu.dk)

Review Timeline:

Submission Date:	3rd Feb 21
Editorial Decision:	4th Mar 21
Revision Received:	27th Apr 21
Editorial Decision:	21st May 21
Revision Received:	28th May 21
Accepted:	9th Jun 21

Editor: Elisabetta Argenzio

Transaction Report:

Thank you for submitting your manuscript entitled "The TOR-dependent phosphoproteome and regulation of cellular protein synthesis" [EMBOJ-2021-107911] to The EMBO Journal. Your study has now been assessed by three reviewers, whose reports are enclosed below for your information.

As you can see, the referees concur with us on the general interest of your findings but also raise several issues that need to be solved before they can support publication in The EMBO Journal. In particular we agree with referee #3 that proving the reproducibility of the experimental setup is a crucial point.

Given the overall interest of your study, we have decided to invite you to submit a new version of the manuscript revised according to the referees' requests. I should add that it is The EMBO Journal policy to allow only a single round of revision, and acceptance of your manuscript will therefore depend on the completeness of your responses in the revised version.

REFeree REPORTS

Referee #1:

Mak et. al., have performed the phosphoproteomic analyses in fission yeast upon TOR inhibition. The authors monitor changes in the phosphoproteome and translation in time-dependent manner upon TOR inhibition and identify proteins that undergo TOR-dependent phosphorylation changes. The authors also claim that TOR regulates protein synthesis in an S6K independent manner in *S. pombe*. Furthermore, the authors show that increased phosphorylation on Tif471 upon TOR inhibition contributes to the repression of translation.

1. The authors claim that the current study "demonstrates the direct impact of TOR-dependent phospho-regulation on the rate of protein synthesis." The TOR-dependent regulation of translation initiation has been shown by many previous studies. Previous studies have not implicated phosphorylation of eIF4G/tif471. However, in this case the phosphorylation is not TOR-dependent but rather TOR-sensitive. The authors should discuss this more openly. In general, they discuss "changes" in phosphorylation without mentioning whether the phosphorylation increases or decreases upon TOR inhibition. Clear separation and analyses of proteins undergoing upregulated or downregulated phosphorylation would be useful.

2. The authors oddly claim that S6K homologs are not needed for the translation repression upon TORC1 inhibition. However, S6Ks are known to have translation-promoting function. Hence, authors must study the translation recovery upon TORC1 activation and determine whether it is dependent or independent of the S6K homologs.

3. The authors show that hyperphosphorylation of Tif471 is necessary for the translation repression upon TOR inhibition. Does hyperphosphorylated Tif471 have less affinity for eIF4E? Is translation recovery upon restimulation with nitrogen faster in Tif471-18A mutant background? More detailed characterization of Tif471 phosphorylation in the regulation of translation is recommended.

4. What is the expression level of Tif471_18A in comparison to wild-type Tif471?

5. Detailed analyses of downregulated phosphosites upon TOR inhibition may help to delineate the potential direct TOR substrates. Defining a consensus TOR target sequences would be useful. Conversely, analysis of the phosphosites whose phosphorylation increases upon TOR inhibition could identify candidate kinases that are activated upon TOR inhibition.

6. The knowledge of TORC1 and TORC2 composition is still ambiguous in *S. pombe*. In the absence of Tor1, Tor2 might contribute to the formation of TORC2. This is supported by the authors' observations that some of the potential targets of TORC2 are affected by Torin treatment in *tor1*^Δ strain. Phospho-proteomic analysis of Ste20^Δ, which is a specific component of the TORC2, would help the authors claim that TORC1 regulates organization of the actin cytoskeleton in *S. pombe*.

7. The authors use terms such as "TOR induced growth inhibition" or "TOR mediated growth inhibition". TOR is generally known as an activator of cell growth.

Referee #2:

This is an elegant, clear and well-written study in which the authors undertake a detailed temporal analysis of the TOR-dependent phospho-proteome in fission yeast. This provides a high resolution analysis of the impact that inhibiting TOR has in global changes in protein phosphorylation and how this links to the regulation of protein synthesis in fission yeast. The quality of this study, the detailed timepoint analysis, use of state-of-the-art mass spectrometry coupled with carefully planned and executed and presented experiments differentiates this study from numerous previous phospho-proteomic analysis of mTOR signalling pathways yeast and mammalian cells.

The analysis and data appear robust. Many interesting and well-studied proteins are identified

whose phosphorylation is impacted by TOR. These include MKK, RAS, NDR, S6K and HIPPO homologues and pathway components as well as regulators of the RHO GTPases. Although the role of these phosphorylation sites is not explored further, the dataset presents a wealth of information that can be exploited by others in future analysis. One of the most important and novel findings in this study is that a protein Tif471, a homologue of eIF4G, appears to lie at the nexus of a new TOR-dependent signalling pathway contributing to protein synthesis. Tif471 is phosphorylated at multiple sites and the authors data provides evidence that TOR mediated phosphorylation of Tif471 contributes to regulating the rate of protein synthesis. I am supportive of this paper being published in the EMBO Journal.

Minor Points

1. Do the authors have any insights into what protein kinases might phosphorylate Tif471?
2. The authors mutate 18 Ser Thr residues on Tif471 to alanine. This is quite a lot and could be predicted to compromise the function of the protein. How do they know that these mutations do not ablate Tif471 function indirectly and the effects they see in protein synthesis is not due to a loss of function mutation in Tif471 rather than a loss of phosphorylation of this enzyme? Is it possible to mutate a lower number of sites to see an effect on protein synthesis?
3. Can the authors undertake a sequence motif analysis of the global Tor regulated sites that they identify in this study? It would be interesting to know types of motifs the Tor regulated phosphorylation sites lie in.

Referee #3:

Mak et al performed a comprehensive study of TOR-dependent (phospho)proteome dynamics in fission yeast. They provide the biggest quantitative phosphoproteome dataset in this organism to date and analyze it in the context of changes in global cellular protein synthesis upon TOR inhibition. They find that the TORC2 complex is dispensable for regulation of protein synthesis and - importantly - provide experimental evidence that the TORC1-mediated phosphorylation of the S6 protein kinases plays no apparent role in the reduction of protein synthesis rates upon TOR inhibition in fission yeast. Instead, they show that a homologue of eIF4G and downstream target of TOR-signalling, Tif471, has a partial role in regulating the rate of protein synthesis. Their study detects numerous other potential TORC1 substrates and provides a simple bioinformatic analysis of the affected molecular processes.

The elegant design of the study and impressive phosphoproteome coverage make this an important resource for a broad scientific audience of the EMBO journal. However, several points need to be strengthened and clarified before acceptance:

- 1) Major: it appears that the proteome and phosphoproteome measurements were done only once for each dataset (no biological replicates are mentioned in the text and the corresponding graphs are devoid of any error bars). Given the complexity of the regulatory network affected by the TOR complex and some obvious "noisiness" of the data (e.g. see fig 4c) it is essential to show that the experimental setup is reproducible. As a minimum, the authors should perform biological replicates in form of short (phospho)proteome measurements, e.g. of selected time points after TOR inhibition, to show that the results correlate well with the "big" datasets and are therefore reproducible.

2) Minor: Detected changes in phosphorylation levels are influenced by protein levels and the changes seen on the phosphorylation level should be normalized with those observed on the proteome level. It is not immediately clear whether this was done.

3) Minor: since the authors detected numerous potential TORC1-substrates, it would be interesting to know whether they detected potential kinase target motifs

We are resubmitting a revised manuscript Mak, Jones and Nurse “The TOR-dependent phosphoproteome and regulation of cellular protein synthesis” (EMBOJ-2021-107911). We thank the referees for their work and comments on our manuscript.

The major concern expressed by the referees and editor was the reproducibility of the phosphoproteomics data, and a request was made to replicate the time course data. In fact, we had provided replicate time courses in the original submission, although it could have been made clearer. The two time courses are not exactly the same, one is for 85 minutes and the second for 40 minutes, but these time courses covered the period when the majority of changes in the phosphoproteome took place. We have now indicated this more clearly in lines 257-259* (PDF; or 255-257 in MS Word) in the manuscript. We have included both the unprocessed and processed data (accounting for unspecific effects of DMSO/Torin1 treatment) in all the data tables (Tables EV2 & EV3) for the 85-minute time course. For the 40-minute time-course experiment, the data is presented for wild type and *tor1Δ* cells in all figures, with all the values compiled also in Tables EV2 & EV3. We have also specifically designed an accompanying data visualisation tool (https://tmak.shinyapps.io/TM_App-Table_EV3/) for ease of access of the data, especially for the subset of phosphosites that show more than 2-fold changes.

For the purpose of the referees, we have compiled replicate graphs of the phosphorylation behaviour of phosphosites in wild type cells. These sites were chosen as they were discussed in the text as showing more than 2-fold phosphorylation changes in either of the datasets (Fig. R1). Note that there are only small variations in the dynamic ranges between the two phosphorylation studies, and the overall kinetic behaviour of the phosphosites is reproducible between the two datasets.

For the rest of this correspondence, we address the remaining questions raised by each of the referees individually.

Referee #1

“Mak et. al., have performed the phosphoproteomic analyses in fission yeast upon TOR inhibition. The authors monitor changes in the phosphoproteome and translation in time-dependent manner upon TOR inhibition and identify proteins that undergo TOR-dependent phosphorylation changes. The authors also claim that TOR regulates protein synthesis in an S6K independent manner in S. pombe. Furthermore, the authors show that increased phosphorylation on Tif471 upon TOR inhibition contributes to the repression of translation.”

1. *The authors claim that the current study "demonstrates the direct impact of TOR-dependent phospho-regulation on the rate of protein synthesis." The TOR-dependent regulation of translation initiation has been shown by many previous studies. Previous studies have not implicated phosphorylation of eIF4G/tif471. However, in this case the phosphorylation is not TOR-dependent but rather TOR-sensitive. The authors should discuss this more openly. In general, they discuss "changes" in phosphorylation without mentioning whether the phosphorylation increases or decreases upon TOR inhibition. Clear separation and analyses of proteins undergoing upregulated or downregulated phosphorylation would be useful.*

Following the referee's comments, we have re-evaluated our usage of the word 'direct' as we agree it could confuse, and have taken it out from the text (9 instances) when describing the relationship between TOR activity and rates of protein synthesis.

However, we disagree with the referee about the use of the word ‘dependent’ in describing the changes in phosphorylation upon TOR inhibition with Torin1, given the evidence using the Torin-resistant mutant as a control that the changes in phosphorylation observed are dependent upon TOR activity.

As proposed by the referee we have separated proteins undergoing up- or downregulated phosphorylations, and have included an additional column in Table EV3 to indicate the direction of change. We performed an analysis to characterise potential motifs amongst the phosphosites, which are described in more detail in later responses (Referee #2 Q3).

2. *The authors oddly claim that S6K homologs are not needed for the translation repression upon TORC1 inhibition. However, S6Ks are known to have translation-promoting function. Hence, authors must study the translation recovery upon TORC1 activation and determine whether it is dependent or independent of the S6K homologs.*

In relation to the role of the S6Ks, we wished to determine if they have a role in modulating the rates of protein synthesis in response to growth inhibitory signals from the TOR pathway, the major objective of our study. Our results provide experimental evidence that ‘the immediate decrease observed in protein synthesis rates upon TOR inhibition is not dependent upon the S6Ks or their downstream targets’ (lines 336-338 in PDF; 332-333 in MS Word). We do not call into question nor discuss whether the S6Ks have translation-promoting function. The suggested experiment on translation recovery upon recovery from TORC1 activation is of interest but would be a major additional study requiring further mass spectrometry analysis, which are already extensive in our study.

3. *The authors show that hyperphosphorylation of Tif471 is necessary for the translation repression upon TOR inhibition. Does hyperphosphorylated Tif471 have less affinity for eIF4E? Is translation recovery upon restimulation with nitrogen faster in Tif471-18A mutant background? More detailed characterization of Tif471 phosphorylation in the regulation of translation is recommended.*

The experiments proposed by the referee might provide mechanistic insights on the functional role of Tif471 phosphorylation and translation repression. However, this is also a major additional study beyond our characterisation of the changes in cellular phosphoproteome upon TOR inhibition in relation to the rates of change in protein synthesis. We consider these suggested experiments are beyond the scope of our study.

4. *What is the expression level of Tif471_18A in comparison to wild-type Tif471?*

Based on a preliminary experiment to characterise other aspects of the *tif471_18A* mutant using Western blotting, we had observed that the expression levels of the Tif471 protein in wildtype and 18A mutant cells were comparable (see Fig. R2). We have now included this observation in the discussion of the manuscript in line 375 (369), as ‘unpublished observations’.

5. *Detailed analyses of downregulated phosphosites upon TOR inhibition may help to delineate the potential direct TOR substrates. Defining a consensus TOR target sequences would be useful. Conversely, analysis of the phosphosites whose phosphorylation increases upon TOR inhibition could identify candidate kinases that are activated upon TOR inhibition.*

See below (Referee #2 Q3).

6. *The knowledge of TORC1 and TORC2 composition is still ambiguous in S. pombe. In the absence of Tor1, Tor2 might contribute to the formation of TORC2. This is supported by the authors' observations that some of the potential targets of TORC2 are affected by Torin treatment in tor1Δ strain. Phospho-proteomic analysis of ste20Δ, which is a specific component of the TORC2, would help the authors claim that TORC1 regulates organization of the actin cytoskeleton in S. pombe*

The referee is suggesting studies on different components of the TORC1 and TORC2 complexes, such as Ste20. This requires a significant amount of additional work, and goes beyond the scope of this study, which focuses on the immediate changes in the phosphoproteome in relation to the change in kinetics of protein synthesis upon TOR inhibition.

7. *The authors use terms such as "TOR induced growth inhibition" or "TOR mediated growth inhibition. TOR is generally known as an activator of cell growth.*

We have adjusted the wording to 'TOR-dependent growth inhibition' in place of 'induced' or 'mediated' for the 2 instances in lines 494 (487) and 499 (492).

Referee #2

"This is an elegant, clear and well-written study in which the authors undertake a detailed temporal analysis of the TOR-dependent phospho-proteome in fission yeast. This provides a high resolution analysis of the impact that inhibiting TOR has in global changes in protein phosphorylation and how this links to the regulation of protein synthesis in fission yeast. The quality of this study, the detailed timepoint analysis, use of state-of-the-art mass spectrometry coupled with carefully planned and executed and presented experiments differentiates this study from numerous previous phospho-proteomic analysis of mTOR signalling pathways yeast and mammalian cells.

The analysis and data appear robust. Many interesting and well-studied proteins are identified whose phosphorylation is impacted by TOR. These include MKK, RAS, NDR, S6K and HIPPO homologues and pathway components as well as regulators of the RHO GTPases. Although the role of these phosphorylation sites is not explored further, the dataset presents a wealth of information that can be exploited by others in future analysis. One of the most important and novel findings in this study is that a protein Tif471, a homologue of eIF4G, appears to lie at the nexus of a new TOR-dependent signalling pathway contributing to protein synthesis. Tif471 is phosphorylated at multiple sites and the authors data provides evidence that TOR mediated phosphorylation of Tif471 contributes to regulating the rate of protein synthesis. I am supportive of this paper being published in the EMBO Journal."

Minor Points

1. *Do the authors have any insights into what protein kinases might phosphorylate Tif471?*

The protein kinase responsible for phosphorylating Tif471 is interesting but we do not have any useful insights about this and do not have any experimental data that is relevant.

2. *The authors mutate 18 Ser Thr residues on Tif471 to alanine. This is quite a lot and could be predicted to compromise the function of the protein. How do they know that these mutations do not ablate Tif471 function indirectly and the effects they see in protein*

synthesis is not due to a loss of function mutation in Tif471 rather than a loss of phosphorylation of this enzyme? Is it possible to mutate a lower number of sites to see an effect on protein synthesis?

The referee expresses concern that the 18 phosphosite mutations on Tif471 may compromise the function of the protein. In Figure EV3, we provide evidence for the unaltered growth rates in the Tif471_18A mutant compared to wild type cells. We have also provided additional experimental evidence for the referees that shows the protein levels of the Tif471_18A-mutant is unaffected compared to wild type cells under steady state growth conditions (Fig. R2; see response for Referees #1 Q4).

In response to the question regarding whether it is possible to mutate less phosphosites and get the same effect on the rate of protein synthesis, we provide for the referees some observations on experiments that we carried out whilst constructing the full 18-alanine mutant. We tested a mutant that had 10-alanine mutations (Tif471_10A), and found that the rate of change in protein synthesis inhibition did not show a notable difference in the residual rates of protein synthesis (Fig. R3).

3. *Can the authors undertake a sequence motif analysis of the global Tor regulated sites that they identify in this study? It would be interesting to know types of motifs the Tor regulated phosphorylation sites lie in.*

All 3 referees asked about potential sequence motifs. Given that the TOR response pathway contains a number of protein kinases, carrying out a motif analysis is not likely to be useful given the different kinases may well have very different substrate motifs. This is borne out by the motif analysis we have provided for the referees using IceLogo of all TOR-dependent sites, as well as TORC1-dependent sites (Fig. R4). It can be seen that the motifs revealed are complex and not easily interpretable, therefore we have not included this analysis in the paper.

Referee #3

“Mak et al performed a comprehensive study of TOR-dependent (phospho)proteome dynamics in fission yeast. They provide the biggest quantitative phosphoproteome dataset in this organism to date and analyze it in the context of changes in global cellular protein synthesis upon TOR inhibition. They find that the TORC2 complex is dispensable for regulation of protein synthesis and - importantly - provide experimental evidence that the TORC1-mediated phosphorylation of the S6 protein kinases plays no apparent role in the reduction of protein synthesis rates upon TOR inhibition in fission yeast. Instead, they show that a homologue of eIF4G and downstream target of TOR-signalling, Tif471, has a partial role in regulating the rate of protein synthesis. Their study detects numerous other potential TORC1 substrates and provides a simple bioinformatic analysis of the affected molecular processes.

The elegant design of the study and impressive phosphoproteome coverage make this an important resource for a broad scientific audience of the EMBO journal. However, several points need to be strengthened and clarified before acceptance:”

1. *Major: it appears that the proteome and phosphoproteome measurements were done only once for each dataset (no biological replicates are mentioned in the text and the corresponding graphs are devoid of any error bars). Given the complexity of the regulatory network affected by the TOR complex and some obvious "noisiness" of the data (e.g. see fig 4c) it is essential to show that the experimental setup is reproducible. As*

a minimum, the authors should perform biological replicates in form of short (phopsho)proteome measurements, e.g. of selected time points after TOR inhibition, to show that the results correlate well with the "big" datasets and are therefore reproducible.

(Discussed in opening paragraphs)

2. *Minor: Detected changes in phosphorylation levels are influenced by protein levels and the changes seen on the phosphorylation level should be normalized with those observed on the proteome level. It is not immediately clear whether this was done.*

We have performed quantitative analysis of the proteome alongside the phosphoproteome and have shown that the overall proteome varies very little compared with the phosphoproteome (Fig. 2A & B). We did not normalise the phosphorylation levels to protein levels but have been careful to check that any of our proteins of interest do not show changes in protein levels that may affect the observed changes in phosphorylation levels.

3. *Minor: since the authors detected numerous potential TORC1-substrates, it would be interesting to know whether they detected potential kinase target motifs*

See response above (Referee #2 Q3)

Thank you again to you and the referees for taking the time to evaluate our manuscript. We hope that this satisfies any concerns, and that our work is now ready for publication.

Yours sincerely,
Tiffany Mak
Andrew Jones
Paul Nurse

N.B. Figures attached below are for the purpose of addressing the referees' responses only, and are denoted with an 'R' before the numbers to distinguish them from the main or extended (EV) figures.

*Two sets of line numbers are provided when referring to specific changes due to the PDF conversion on the submission portal resulting in a change in margin size, and hence line numbers, of the submitted MS Word document. The first numbers refers to the numbering on the converted (and merged) PDF documents, and the numbers in brackets are for the unconverted MS Word format as detailed in the first two instances.

Figure R1

Figure R1 – Phosphorylation kinetics of example phosphosites in wild type cells

Representative graphs of the phosphosites mentioned in the main text with more than 2-fold phosphorylation change. Dark green indicates the unprocessed data for the 85-minute time course in wild type cells, and light green for the 40-minute time course, overlaid on the same respective graph in all panels for **(A)** Ras1, **(B)** Not2, **(C)** Igo1, **(D)** Ccr4, **(E)** Sck1 and **(F)** Tif471. Position of phosphosites indicated in titles after the protein names and the M number in brackets represent the multiplicity.

Figure R2

Figure R2 – Protein expression levels for the Tif471 protein in wild type and *tif471_18A* mutant cells

Western blot showing the changes in relative protein expression levels over time of Torin1 treatment in hours indicated on top of each lane, for Tif471-tagged wild type cells in lanes 1-3, and *tif471_18A* mutant in lanes 4-6. Both the wild type and mutant forms of the Tif471 protein were tagged with the v5-epitope in both strains and detected with mouse monoclonal anti-V5 antibody (AbD seroTEC, Cat#MCA1360; RRID: AB_322378). Tubulin was used as the loading control, and was detected using the mouse monoclonal anti-alpha tubulin antibody (TAT1; (Woods *et al*, 1989)). This experiment was carried out for another purpose, but the T=0 minutes timepoint for the respective strains are relevant to the referees' questions.

Figure R3

Figure R3 – Rates of inhibition for protein synthesis in the *tif471_10A* mutant in relation to wild type cells

Graph showing the changes in rates of protein synthesis of the *tif471_10A* mutant (S912A, S919A, S921A, S1293A, S1300A, S1333A, S1349A, T1351A, S1353A & S1357A) compared to the wild type control (*tif471*⁺) upon Torin1 (5 μM) treatment. The two graphs illustrate the same result with the one on the left showing untransformed data and the log₂ transformed version on the right.

Figure R4

Figure R4 – Motif analysis for general TOR or TORC1-dependent phosphosites

Motif analysis performed using IcelLogo (Colaert *et al*, 2009) on TOR-dependent phosphosites.

A – B Analysis of all TOR-dependent phosphosites that showed phosphorylation changes of more than 2-fold within 40 minutes in the increase (**A**) or decrease (**B**) directions for both studies combined. All detected phosphosites from the two studies were combined and used as the reference set for the analysis.

C – D Analysis of TORC1-dependent phosphosites that exhibited phosphorylation changes of more than 2-fold from the 40-minute time course study. Sites that showed either an increase (**C**) or decrease (**D**) in phosphorylation within 40 minutes were compared against a reference dataset of all the phosphosites detected from the 40-minute time-course study.

REFERENCES

Colaert N, Helsens K, Martens L, Vandekerckhove J, Gevaert K (2009) Improved visualization of protein consensus sequences by iceLogo. *Nat Methods* 6: 786-787

Woods A, Sherwin T, Sasse R, MacRae TH, Baines AJ, Gull K (1989) Definition of individual components within the cytoskeleton of *Trypanosoma brucei* by a library of monoclonal antibodies. *J Cell Sci* 93 (Pt 3): 491-500

Thank you for submitting your revised study. The manuscript has now been sent back to referee #1 and #3, whose comments are appended below.

As you will see, reviewer #3 finds that his/her criticisms have not been satisfactorily addressed. Conversely, while referee #1 appreciates the quality of the study, s/he also stresses that the requested follow-up characterization of the two phospho-proteomic data sets has not been performed.

Given the interest in your phospho-proteomic study as a resource article, we would pursue publication of your study in The EMBO Journal. However, I would ask you to emphasize the "resource" character of your work in the abstract and discussion sections of the manuscript, as well as to tone down the claims related to the functional part of the study.

In addition, there are few editorial issues concerning the text and the figures that I need you to address before we can officially accept your manuscript.

REFEREE REPORTS

Referee #1:

The authors have not performed further experiments and have thus not satisfactorily addressed our concerns. Our suggestions were intended to improve the manuscript. The current version is merely a description of two phospho-proteomic data sets without much follow-up characterization. However, since the other two reviewers were enthusiastic about the study, we leave the decision to the editor. What is described in the manuscript is certainly of publication quality.

Referee #3:

The authors have addressed all points I raised in my initial report. Although identical replicates would be better (e.g. to address statistical significance) I agree that the two experiments (time

courses of 40 and 85 min) can be considered as independent replicates. This is now also better explained in the text. The authors also addressed most of the other reviewer's comments and the manuscript has considerably improved. Since the findings have a strong resource character, I agree with the authors that extensive additional experiments to clarify the exact role of Tif471 phosphorylation in repression of translation are beyond the scope of the manuscript. I recommend acceptance of the manuscript in its present form.

Thank you for your favourable response to pursue the publication of our study in The EMBO Journal. Please find below a point-by-point address to the comments below (highlighted in blue), along with the resubmitted files (EMBOJ-2021-107911R) on the submission portal:

Given the interest in your phospho-proteomic study as a resource article, we would pursue publication of your study in The EMBO Journal. However, I would ask you to emphasize the "resource" character of your work in the abstract and discussion sections of the manuscript, as well as to tone down the claims related to the functional part of the study.

We have now altered the texts in the abstract (lines 17, 21-25) and discussion (lines 409, 410, 497, 498, 503 & 504) sections of the manuscript to emphasise the 'resource' character. We have also altered the sentences describing the functional work referring to conclusions well supported by our experimental data.

Referee #1:

The authors have not performed further experiments and have thus not satisfactorily addressed our concerns. Our suggestions were intended to improve the manuscript. The current version is merely a description of two phospho-proteomic data sets without much follow-up characterization. However, since the other two reviewers were enthusiastic about the study, we leave the decision to the editor. What is described in the manuscript is certainly of publication quality.

Referee #3:

The authors have addressed all points I raised in my initial report. Although identical replicates would be better (e.g. to address statistical significance) I agree that the two experiments (time courses of 40 and 85 min) can be considered as independent replicates. This is now also better explained in the text. The authors also addressed most of the other reviewer's comments and the manuscript has considerably improved. Since the findings have a strong resource character, I agree with the authors that extensive additional experiments to clarify the exact role of Tif471 phosphorylation in repression of translation are beyond the scope of the manuscript. I recommend acceptance of the manuscript in its present form.

I am pleased to inform you that your manuscript has been accepted for publication in The EMBO Journal.

Corresponding Author Name: Tiffany Mak

Manuscript Number: 2021-107911